## Registered report

behaviour/psychology

observability, prosocial behaviour, conservation (ecological behaviour), pro-environmental behaviour, signalling, pro-environmental behaviour task

**Author for correspondence:**
Florian Lange
e-mail: florian.lange@kuleuven.be

# Green when seen? No support for an effect of observability on environmental conservation in the laboratory: a registered report

Florian Lange[1], Cameron Brick[2,3] and Siegfried Dewitte[1]

[1]Behavioral Engineering Research Group, KU Leuven, Leuven, Belgium
[2]Department of Psychology, University of Amsterdam, Amsterdam, The Netherlands
[3]Department of Psychology, University of Cambridge, Cambridge, UK

 FL, 0000-0002-8336-5608; CB, 0000-0002-7174-8193

Understanding how humans navigate the tension between selfish and prosocial behaviour is central to addressing social dilemmas and several environmental issues. Many accounts predict that human prosociality would increase in the presence of observing individuals. Previous studies on this observability effect predominantly relied on artificial observability manipulations and low-cost measures of prosociality. In the present Registered Report, we used a recently validated laboratory procedure of repeated dilemmas to test whether the presence of actual observers affects costly prosocial behaviour in the domain of environmental conservation. When completing this dilemma task, participants repeatedly chose between minimizing the length of the laboratory session and minimising wasted energy from a bank of LED lights. Their choices were made either in private or in the presence of actual observers. Contrary to our expectation, we did not observe higher rates of energy-conserving behaviour when participants' choices were being observed. Manipulation and robustness checks indicate that this lack of a finding is unlikely to be owing to arbitrary methodological choices. In view of these findings, we argue that a more comprehensive analysis of situation- and behaviour-specific consequences might be necessary to predict how particular behaviours are affected by observability.

# 1. Introduction

Humans engage in a variety of behaviours that benefit the people around them. They spend their spare time volunteering [1], donate money [2] and blood [3], help others cross the street [4], and share their resources with anonymous strangers in economic dilemma tasks [5]. Sometimes, the beneficial consequences of their behaviour even extend across borders and generations. When showing pro-environmental behaviour (i.e. behaviour that benefits the natural environment [6]), individuals essentially cooperate with all humanity and other living beings, present and future [7,8]. In view of global environmental deterioration and climate change [9,10], such behaviour is critically needed. Understanding the factors that give rise to pro-environmental behaviour is an important challenge for the behavioural sciences.

When the pro-environmental consequences of a behaviour are aligned with benefits for the individual, the occurrence of pro-environmental behaviour may not be much of a conundrum. Where a destination can be reached by bicycle more quickly than by car, and where switching off devices reduces carbon emissions and utility bills alike, the assumption of narrow self-interest is sufficient to account for pro-environmental behaviour. In many cases, however, environmental benefits are accompanied by a non-trivial cost imposed on the behaving individual [11,12]. Electric cars and organic food sell at a price premium, waste separation requires time, and the scholar refusing air travel misses out on inspiring conferences overseas. Why are people willing to incur these costs?

One possibility to explain costly pro-environmental behaviour refers to the social consequences of such a behaviour [13]. Individuals may behave pro-environmentally simply because other people are likely to respond favourably to individuals who behave in such a way. When a group classifies a certain behaviour as good, group members showing this behaviour are rewarded by acts of approval and affection [14, pp. 323–326]. Pro-environmental actions are widely considered good behaviour. More than 90% of all citizens of the European Union report that protecting the environment is of personal importance to them [15]. Some scholars even consider environmental protection 'one of the most established norms within world society' [16, p. 317]. Recent evidence suggests that individuals who comply with this norm are viewed to be more prosocial, attractive, respected, and sophisticated [17,18, but see 19]. In addition, participants who behaved more pro-environmentally appear to be treated more favourably in social interactions in some studies [20], but not all research confirms this pattern [21].

Are such consequences sufficient to bring pro-environmental behaviour under the (partial) control of social norms? The success of interventions conveying normative information suggests that they are. Individuals show more pro-environmental behaviour when being informed that a majority of others approve of this behaviour or show it themselves [22–27, see also 28]. In other words, pro-environmental behaviour is more likely to occur when it is made salient to individuals that their social environment is likely to reward such behaviour.

A second prediction of a social norm account of pro-environmental behaviour relates to the role of behavioural observability. If pro-environmental behaviour is indeed driven by social consequences, it should be more prevalent in the presence versus absence of potential observers [29]. When pro-environmental behaviour is observed, it can lead to those social consequences that reinforce its occurrence. When their behaviour is observed by others, individuals can be held accountable for it: they may enjoy social benefits for incurring personal cost to benefit a greater good or they may be asked to justify why they did not do so.

## 1.1. Observability effects on prosocial and pro-environmental behaviour

A recent meta-analysis revealed a small effect of observability in the broad domain of prosocial behaviour [30]. This effect was qualified by a number of moderating variables, many of which relate to the operational definition of observability. For example, observability seemed to exert a substantially stronger effect on prosocial behaviour when participants were exposed to the scrutiny of actual observers rather than to artificial cues of being watched (i.e. images of watching eyes). In the present study, we consider behaviour to be observable when the target individuals as well as their behaviour can be physically observed by others. Note that this definition only requires that a behaviour can be observed, not that every instance of this behaviour has to be observed. Similarly, it does not require that individuals are made explicitly aware of being observed.

Some authors have reported evidence suggestive of an observability effect in the more specific domain of pro-environmental behaviour [13,31–33]. However, none of these studies involved

manipulations of actual observability as defined above. Authors either studied the effect of an artificial observation cue [31,32], asked their participants to imagine shopping in a private versus public setting [13], or informed their participants that their decisions (made in private) would be revealed to other participants after the study [33].

This lack of conclusive tests of the observability effect is a critical methodological limitation in contemporary research on pro-environmental behaviour. Until recently, no validated protocol for the measurement of pro-environmental behaviour was available. Pro-environmental behaviour had to be studied in the field (where controlled manipulations of observability are hardly possible; e.g. [31]), in hypothetical self-report scenarios (e.g. [13]), or in single-trial *ad hoc* tasks in the laboratory (e.g. [33]). The valid study of observability effects on pro-environmental behaviour would require a procedure that (i) is psychometrically established, (ii) elicits a visible type of pro-environmental behaviour and (iii) allows manipulating the degree to which other individuals can observe this kind of behaviour.

## 1.2. The pro-environmental behaviour task

Lange *et al.* recently designed and validated a procedure that meets the criteria listed above: the pro-environmental behaviour task (PEBT) [34]. On each of the trials of the PEBT, participants decide whether they want to use the car or the bicycle for a simulated trip. Their choices relate to actual consequences for themselves and for the environment. Choosing the bicycle over the car prolongs the time participants have to wait until the next trial starts (and thus the time participants spend on the task). However, choosing the car over the bicycle turns on special lights on the desk (and thus wastes energy and emits greenhouse gases). Over multiple trials, participants must balance their waiting time cost and their ecological footprint.

Psychometric studies revealed that the proportion of bicycle choices on the PEBT can serve as a valid, objective measure of actual pro-environmental behaviour. It is affected by variables that should theoretically affect pro-environmental behaviour (such as individual cost and environmental benefits) and related to variables that should theoretically relate to pro-environmental behaviour (such as environmental attitudes, concern, values and identity). In addition, the proportion of pro-environmental PEBT choices has repeatedly been shown to correlate with self-reports of pro-environmental behaviour in everyday life [34,35]. Critically, this measure can be obtained under controlled laboratory conditions where factors such as observability can be manipulated. An individual sitting next to a PEBT participant can observe how this participant uses the computer mouse to choose one of the two options and whether this behaviour illuminates the PEBT lights or not. If the view of this observing participant is occluded (e.g. by a divider), neither the choice behaviour nor its environmental consequences can be observed by others. The PEBT thus allows studying the effect of the presence of an actual observer on a valid measure of actual pro-environmental behaviour. It is this potential that allowed for a robust test of the observability effect on pro-environmental behaviour in the present study. By this means, we aimed to contribute to a broader understanding of how social factors shape human prosociality.

## 1.3. Hypothesis

The study described in the following tested the effect of manipulating actual observability on an objective and consequential measure of pro-environmental behaviour. Participants in the observable condition of our design were expected to show a larger proportion of environmentally friendly choices on the PEBT as opposed to participants in the non-observable condition.

# 2. Methods

## 2.1. Power analysis and sample size rationale

We aimed to recruit 176 target participants (henceforth targets) and 176 additional observers. This sample size allowed detecting an observability effect of $d = 0.50$ with *a priori* power of 95% (given $\alpha = 0.05$, one-sided). Our effect size estimate of $d = 0.50$ was based on a review of the scarce literature on observability effects on pro-environmental behaviour. The effect of eyes images on littering behaviour ranged between $d = 0.59$ and $d = 0.83$ in the studies by Bateson *et al.* [31], while being smaller ($d = 0.37$) in the study by Ernest-Jones *et al.* [32]. Griskevicius *et al.* found a large effect ($d = 0.73$) of asking participants to imagine shopping in a store setting versus online setting on the preference for green products (but only when

participants read a story supposed to activate status motives, [13]). Similarly, Vesely & Klöckner [33] found a large observability effect ($d = 0.78$) on pro-environmental donations (but only when participants were given information indicating that making a large donation was the norm). The last study might be the study most similar to our own as it also involved measurement of actual pro-environmental behaviour in the laboratory. As this study had a relatively small sample size ($n = 34$ per cell), we reasoned that the associated effect size might be an optimistic estimate of the true effect of observability. We thus decided to base our sample size rationale on a slightly smaller effect-size estimate of $d = 0.50$. Power analysis (G*Power 3.1.9.2, [36]) showed that $n = 88$ participants per group are necessary to detect group differences of $d = 0.50$ or larger with a statistical power of 95% using a one-sided independent-samples $t$-test.

## 2.2. Recruitment

Participants received 6 euros for a study that took about 30 min. The study was advertised to potential participants enlisted in the faculty's subject pool. We began by offering 352 testing slots. Additional slots were opened until the target sample size was reached. Owing to conservative overbooking, we considered the possibility that the target sample size might be slightly exceeded (i.e. by a maximum of 12 participants). For this case, we preregistered to keep the data from additional participants. Participants were combined into target-observer pairs. Up to four target-observer pairs were tested in each session. Registration for each particular session was open to either only male participants or only female participants, thus ensuring that all observers were of the same gender as their corresponding targets. Before a testing session, the experimenter checked the number of participants who had signed up for the session. If this number was uneven, one of the participants was tried to be shifted to another session before coming to the laboratory. Nonetheless, last-minute cancellations led to an uneven number of participants in some testing sessions. In this case, one of the participants could not be paired with an observer. This participant completed the target procedure (see below), but the resulting data was excluded from all confirmatory analyses and the participant was not counted towards the total sample size.

In the email advertising the study, participants were informed that a proficient level of English was required for participation in this study. Participants were excluded if they did not follow the study instructions; that is, if they refused to complete the laboratory tasks or to hand over their phones for the time of the experiment.

## 2.3. Procedure

Upon arrival at the laboratory, participants were instructed to leave their belongings (especially their phones) in the front area of the laboratory before they were shown to their testing cubicle. Participants were randomly assigned to a participant role, with half becoming observers and half becoming targets. In addition, each pair of target and observer was assigned to one of the two conditions of the between-subject factor observability (observable versus non-observable). To this end, participants drew lots before entering the laboratory. As soon as a participant had completed all tasks, she or he left the laboratory, irrespective of whether the other participant in the corresponding target-observer pair had already finished.

## 2.4. Participant roles

In the online invitation for this study as well as on the informed consent form, participants were told that we were running two different studies in parallel and that we decided to combine these two studies for efficiency. Participants received general information about both studies and were informed that they would be assigned to one of the studies upon their arrival at the laboratory. They were told that one of the studies involved a music evaluation task and that the other study involved a computer task for the measurement of pro-environmental behaviour (i.e. the PEBT, see below). They further learned that choosing the environmentally unfriendly option on this task would turn on some lights that consume unnecessary energy. Therefore, observers knew how to interpret the behaviour of the target, and targets knew that observers could interpret their behaviour on the PEBT as environmentally friendly or unfriendly.

### 2.4.1. Targets

Targets completed two blocks of 20 trials on the PEBT [34], a laboratory task that measures objective pro-environmental behaviour and that is implemented in OpenSesame [37]. It involved a series of choices between two response options. On each choice trial, participants had to decide whether they wanted

**Table 1.** Waiting times associated with the two PEBT options in seconds. (PEBT, pro-environmental behaviour task; WTD, waiting time difference.)

| car | bicycle | difference (WTD) |
|---|---|---|
| 5 | 15 | 10 |
| 10 | 20 | |
| 15 | 25 | |
| 20 | 30 | |
| 5 | 20 | 15 |
| 10 | 25 | |
| 15 | 30 | |
| 20 | 35 | |
| 5 | 25 | 20 |
| 10 | 30 | |
| 15 | 35 | |
| 20 | 40 | |
| 5 | 30 | 25 |
| 10 | 35 | |
| 15 | 40 | |
| 20 | 45 | |
| 5 | 35 | 30 |
| 10 | 40 | |
| 15 | 45 | |
| 20 | 50 | |

to use the car or the bicycle for a particular trip. The task could also be run with neutral, connotation-free labels. Validation studies have not found participants' behaviour on the task or its psychometric properties to be affected by the choice of labels [34]. Participants directly experienced two different consequences of their choice and they were explicitly informed that 'the choices [they] make have consequences for [themselves] (that is, they determine how long the experiment takes) as well as for the environment (that is, they determine how much energy is consumed during the experiment).' First, following their choice, participants had to endure a waiting period before they could choose a mode of transportation for the next trip. The waiting time for the bicycle option was always longer than the waiting time for the car option. Before making their choice, participants were explicitly informed about the waiting periods associated with the two options as well as about the waiting time difference (WTD) between the two options. The WTD factor indicates the difficulty or cost of showing pro-environmental behaviour (i.e. choosing the bicycle option) on the PEBT. The WTD was varied randomly at five levels (10 s, 15 s, 20 s, 25 s and 30 s). These levels were based on prior studies [34,35] finding that these WTDs produce largely non-skewed distributions in overall choice. Waiting times and WTDs for the PEBT trials are displayed in table 1.

Second, every time the car was chosen, an array of USB-powered lights located on the desk of the target was illuminated for the duration of the trip. In one of the two PEBT blocks, four lights were illuminated, and in the other block, 12 lights were illuminated. The order of the blocks was counterbalanced across participants. At the beginning of each block as well as on every trial, participants were informed about the number of lights that would be turned on by choosing the car option. They were also informed about the approximate amount of $CO_2$ emissions produced by powering the lights (i.e. 3000 mg hr$^{-1}$ in the four-lights block and 9000 mg hr$^{-1}$ in the 12-lights block). Between blocks, targets were explicitly informed about the change in the number of lights illuminated and the amount of $CO_2$ emissions produced by choosing the car option. This change in energy consumption was further highlighted with the words 'For the rest of this task, the car option is less [more] energy-efficient than before. That means that choosing the car will consume a larger [smaller] amount of energy'.

Depending on participants' choices, the current version of the PEBT took 10–25 min to complete. After the final trial of the second block, targets were asked to close the task and to proceed by completing a short questionnaire on the computer. The questionnaire contained demographic questions regarding participants' age, gender, current profession and native language as well as the possession of a bicycle, a car and a driver's license. Participants' responses to these questions were used for the description of the sample. We also assessed a number of exploratory variables including (i) donations to an environmental organization, (ii) prosocial donations made to the observer, and (iii) environmentalist identity [36,38]. Finally, targets were asked to rate the testing situation in our laboratory on the dimensions temperature (1, too cold – 7, too hot), lighting conditions (1, too dark – 7, too bright) and privacy/anonymity (1, very low – 7, very high). Responses to the last item were used as a quasi-manipulation check (see below).

### 2.4.2. Observers

Observers completed a control task of rating music on the computer, also implemented in OpenSesame. They were presented with the first 60 s of 15 contemporary popular music pieces. While observers were listening to the pieces via headphones, the screen on their desk remained blank. As a corollary, no demands were placed on observers' visual resources during music presentation. This design feature should increase the probability that observers direct their visual attention towards locations other than their screen (e.g. towards the target sitting next to them). After the music stopped, observers were presented with three rating scales on the screen. They were asked 'How often have you heard this song before?' (1, never – 10, very often), 'How much did you like this song?' (1, not at all – 10, very much), and 'How much do you regret that the song stopped playing?' (1, not at all – 10, very much). In addition, they responded to the question 'How did this song make you feel?' on 5-point pictorial assessment scales of valence, arousal, and dominance [39]. Completing this task took about 25 min. This length was chosen to ascertain that observers were present for the entire time targets spent on the PEBT.

After the final music rating, observers were instructed to close the task and complete a short questionnaire on the computer. Prosocial donations made to the target and observers' belief about the amount the target donated to them were assessed as exploratory variables. Observers then indicated how much attention they had spent on the behaviour of the target (1, none – 7, very much), how well they could see what the target was doing during the session (1, not at all – 7, very well), and whether they knew the target outside the laboratory (yes – no). These data were used for additional robustness and manipulation checks. The questionnaire ended with the environmentalist identity scale (exploratory variable) as well as demographic questions regarding observers' age, gender, current profession, native language and music preferences.

## 2.5. Observability manipulation

Four pairs of targets and observers were tested simultaneously in a laboratory room that was partitioned into 10 partially enclosed cubicles. Members of a pair were tested in adjacent cubicles. Adjacent cubicles were separated by moveable walls. Before each session, the walls separating the members of two of the target-observer pairs were removed (figure 1). As a consequence, two of the observers had an unobstructed view of their respective target (and no other participant). They were also able to see the screen and desk of the target including the lights that were turned on by choosing the environmentally unfriendly PEBT option. These target-observer pairs were in the observable condition. The other two observers were not able to see their respective target. They and their targets were in the non-observable condition. For half of the sessions, the cubicles in the front area of the laboratory (i.e. close to the door) were used to create the observable condition, and for the other half, the observable condition was created in the back area (i.e. close to the window).

## 2.6. Analysis

To test our hypothesis, we compared the mean proportion of bicycle choices made by targets in the observable condition to the mean proportion of bicycle choices made by targets in the non-observable condition. Following recent recommendations [40], we did not make our test choice contingent on the results of assumption checks, but instead used Welch's $t$-test by default. The level of significance was set to $\alpha = 0.05$ (one-sided). Test results are accompanied by a measure of effect size (Cohen's $d$, with the pooled s.d. weighted for sample sizes) and its corresponding 95% confidence interval (CI). We inspected the data from our dependent measure for possible floor or ceiling effects, but given the

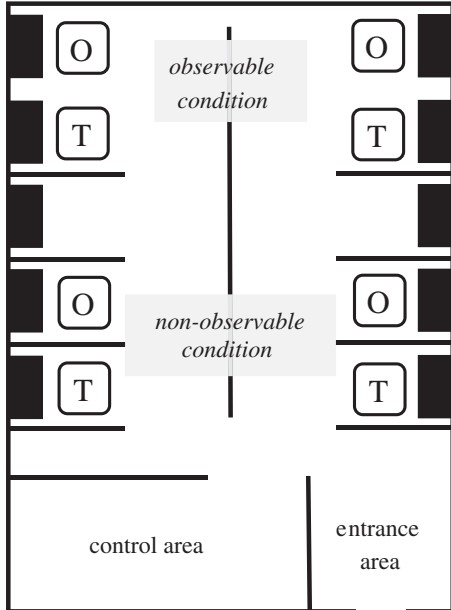

**Figure 1.** Illustration of how observability was manipulated in the testing laboratory.

extensive pre-testing of the PEBT, such effects seemed unlikely to occur. For the confirmatory test of the observability effect, we did not apply any data-based criteria for the exclusion of participants. For the case of finding a significant effect of observability in the predicted direction, we preregistered a robustness check of repeating the above analysis while excluding participants who indicated that they knew each other outside of the laboratory.

The two groups were also compared on three (quasi-)manipulation check variables. Observers' ratings of their ability to see what the target was doing were expected to be higher in the observable than non-observable condition. These ratings were compared using a Welch's $t$-test at $\alpha = 0.05$ (one-sided). We consider this comparison to be a manipulation check because it closely corresponds to our operational definition of observability (see above). By contrast, observers' ratings of how much attention they spent on the target's behaviour and targets' ratings of the privacy/anonymity of the testing situation are regarded as quasi-manipulation checks. As defined above, observability does not necessarily involve the actual act of observing or reductions in perceived privacy. However, both variables are likely to be affected by a successful manipulation of observability and the extent to which they were affected by our manipulation of observability informs the interpretation of our results. Therefore, we descriptively compared ratings of attention and perceived privacy between the observable and non-observable condition and report the corresponding effect sizes and 95% CIs.

## 3. Results

A total of 204 target participants completed the PEBT, 27 of which were not paired with an observer in the adjacent cubicle. One participant in the non-observable condition completed only one of the two PEBT blocks owing to a software crash. Because the data from this participant cannot be compared with the rest of the sample, they were excluded from all analyses. Of the 176 target participants included in the main analyses (111 female, 64 male, one preferred not to say; $M_{\mathrm{age}} = 23.65$, s.d.$_{\mathrm{age}} = 4.91$), 92 were assigned to the observable condition and 84 were assigned to the non-observable condition. Table 2 displays sample characteristics (based on targets' self-reports), separately for the observable and non-observable condition.

Observers rated the behaviour of the participant next to them (i.e. the corresponding target) to be more visible in the observable ($M = 4.00$, s.d. = 1.96) than in the non-observable condition ($M = 1.20$, s.d. = 0.92): $t_{129.89} = 12.27$, $p < 0.001$, $d = 1.81$, 95% CI (1.45, 2.16). They also reported paying more attention to the target's behaviour in the observable ($M = 2.48$, s.d. = 1.23) compared to the non-observable condition ($M = 1.17$, s.d. = 0.62): $d = 1.18$, 95% CI (0.86, 1.50). These analyses are based on $n = 175$ participants as one observer

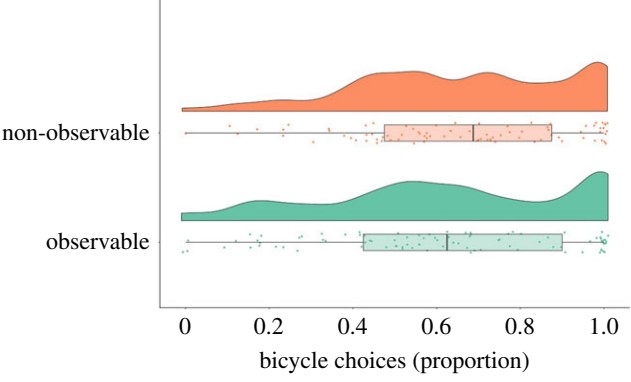

**Figure 2.** The proportion of bicycle choices on the pro-environmental behaviour task (PEBT) by observability condition as a raincloud plot [41]. Boxes indicate interquartile ranges. Vertical black lines are medians.

**Table 2.** Sample characteristics of target participants in the observable and non-observable condition. (CI, confidence interval; OR, odds ratio.)

|  | observable | non-observable | effect size (95% CI) |
|---|---|---|---|
| female participants | 65% | 61% | OR=0.82 (0.45, 1.52) |
| students | 87% | 90% | OR=1.43 (0.55, 3.68) |
| Dutch native speakers | 49% | 33% | OR=0.52 (0.28, 0.96) |
| driver's licence | 64% | 61% | OR=0.86 (0.47, 1.59) |
| car possession | 20% | 24% | OR=1.29 (0.63, 2.64) |
| bicycle possession | 83% | 79% | OR=0.77 (0.37, 1.63) |
| prior PEBT experience | 21% | 15% | OR=0.70 (0.32, 1.53) |
| age (years) | $M = 23.49$, s.d. $= 4.91$ | $M = 23.83$, s.d. $= 4.93$ | $d = 0.07$ (−0.23, 0.37) |
| environmentalist identity | $M = 15.93$, s.d. $= 5.72$ | $M = 16.10$, s.d. $= 4.68$ | $d = 0.03$ (−0.26, 0.33) |

did not complete the questionnaire. In addition, targets rated the laboratory conditions to be less private/anonymous in the observable ($M = 4.11$, s.d. $= 1.61$) versus non-observable condition ($M = 5.61$, s.d. $= 1.16$): $d = −1.06$, 95% CI (−1.38, −0.75). These results indicate that the insertion versus removal of dividers between targets and observers was successful in manipulating relative observability of targets' behaviour on the PEBT.

Figure 2 displays the frequency of pro-environmental behaviour as a function of experimental condition. The proportion of bicycle choices on the PEBT was not larger in the observable ($M = 61.77\%$, s.d. $= 29.40\%$) than in the non-observable condition ($M = 65.92\%$, s.d. $= 25.54\%$): $t_{173.59} = −1.00$, $p = 0.84$, $d = −0.15$, 95% CI (−0.45, 0.15). In the following, we will describe a series of exploratory analyses that might inform the interpretation of this result. It should be noted that, for a wide range of plausible effect-size estimates, many of these exploratory analyses are likely to be underpowered.

## 3.1. Exploratory analyses

### 3.1.1. Robustness checks

Software problems caused two targets (one observable, one non-observable) to complete the questionnaire before the PEBT. One additional target (non-observable) was erroneously paired with an observer of the opposite gender. These three cases were included in the main analyses, but excluded for a robustness analysis. In this robustness analysis, the proportion of bicycle choices on the PEBT was not larger in the observable than in the non-observable condition: $t_{170.84} = −0.90$, $p = 0.82$, $d = −0.14$, 95% CI (−0.44, 0.16).

**Table 3.** Pearson correlation coefficients between PEBT performance and potential covariates. (PEBT, pro-environmental behavior task; obs., observer.)

| | proportion of bicycle choices | | | mean response time | | |
|---|---|---|---|---|---|---|
| | total | observable | non-observable | total | observable | non-observable |
| mean response time | 0.24 | 0.23 | 0.22 | | | |
| rated privacy (target) | 0.20 | 0.26 | 0.05 | 0.29 | 0.33 | 0.08 |
| rated visibility (obs.) | −0.02 | 0.13 | −0.13 | −0.21 | −0.14 | 0.00 |
| rated attention (obs.) | −0.06 | −0.06 | 0.08 | −0.07 | 0.07 | 0.07 |
| gender (female = 1, other = 0) | 0.30 | 0.40 | 0.19 | 0.04 | 0.08 | 0.01 |
| job status (student = 1, other = 0) | 0.04 | 0.07 | −0.02 | 0.05 | 0.21 | −0.16 |
| first language (Dutch = 1, other = 0) | −0.12 | −0.09 | −0.13 | −0.27 | −0.23 | −0.26 |
| driver's licence | −0.07 | −0.19 | 0.07 | −0.05 | 0.04 | −0.12 |
| car possession | −0.08 | −0.16 | 0.00 | 0.05 | 0.00 | 0.09 |
| bicycle possession | −0.04 | −0.05 | −0.01 | −0.08 | −0.07 | −0.07 |
| prior PEBT experience | −0.17 | −0.13 | −0.23 | −0.35 | −0.37 | −0.31 |
| age | −0.08 | −0.15 | 0.02 | −0.01 | −0.17 | 0.14 |
| environmentalist identity | 0.29 | 0.33 | 0.23 | 0.06 | 0.02 | 0.10 |

In addition, observers reported knowing four targets in the observable condition and one target in the non-observable condition. While we preregistered running a robustness check excluding these targets only for the case that we would find support for our hypothesis, we later reasoned that this robustness check would be informative regardless. Results of the one-sided Welch's $t$-test comparing the proportion of bicycle choices between conditions were similar when excluding these five targets: $t_{167.64} = -1.10$, $p = 0.86$, $d = -0.17$, 95% CI (−0.47, 0.13).

We also tested whether the observed mean difference between the observable and non-observable condition (i.e. $d = 0.15$) would change when adding any of the variables listed in table 3 as a covariate. The resulting effect sizes (based on estimated marginal means adjusted for covariates) ranged between $d = -0.22$ and 0.04. None of these effect sizes support a positive effect of observability on pro-environmental PEBT choices.

### 3.1.2. Secondary outcome measure

After the PEBT, targets were given the opportunity to donate up to 2 euros of their participation fee to an environmental organization. Targets in the observable condition donated an average of €0.51 (s.d. = 0.74), compared to €0.52 (s.d. = 0.71) in the non-observable condition: $d = -0.01$, 95% CI (−0.31, 0.28). In accordance with what we told our participants, the sum of targets' donations was multiplied by 1.5 and donated to an organization that offsets carbon emissions.

Targets were also given the opportunity to donate up to 2 euros of their fee to the corresponding observer and vice versa. On average, targets donated €0.46 (s.d. = 0.65) in the observable and €0.48 (s.d. = 0.67) in the non-observable condition: $d = -0.03$, 95% CI (−0.33, 0.27), whereas observers donated €0.90 (s.d. = 0.71) in the observable and €0.97 (s.d. = 0.71) in the non-observable condition: $d = -0.10$, 95% CI (−0.40, 0.20). Targets said they expected to receive €0.35 (s.d. = 0.56) from the corresponding observer in the observable condition and €0.33 (s.d. = 0.52) in the non-observable condition: $d = 0.04$, 95% CI (−0.26, 0.33). Observers expected to receive €0.74 (s.d. = 0.66) from the target in the observable condition and €0.76 (s.d. = 0.60) in the non-observable condition: $d = -0.03$, 95% CI (−0.33, 0.27). Taken together, these results do not provide support for an effect of our observability manipulation on any of the donation-related variables listed above. Note that all these variables were multimodally distributed, with peaks at €0, €0.50, €1 and €2.

### 3.1.3. Social consequences of pro-environmental behaviour

We calculated Pearson correlation coefficients to explore whether more pro-environmental behaviour was related to advantages in the interpersonal donation game. Only small correlations were observed between observers' donations to the target and targets' pro-environmental behaviour, observable condition: $r = 0.11$, 95% CI ($-0.10$, 0.31), non-observable condition: $r = 0.11$, 95% CI ($-0.11$, 0.32). The correlation between observers' expectancies of targets' donations and targets behaviour on the PEBT was $r = 0.16$, 95% CI ($-0.05$, 0.35) in the observable condition and $r = -0.04$, 95% CI ($-0.25$, 0.18) in the non-observable condition. Hence, our data do not suggest that being observed showing pro-environmental behaviour relates to reputational benefits.

### 3.1.4. Potential moderators of the observability effect

Targets' environmentalist identity was positively correlated to their proportion of bicycle choices on the PEBT: $r = 0.29$, 95% CI (0.15, 0.42), but environmentalist identity did not moderate the effect of observability on PEBT behaviour: $b = 0.00$, 95% CI ($-0.02$, 0.01).

Both the individual cost of choosing the bicycle option (i.e. the WTD between bicycle and car): $F_{2.18, 379.49} = 210.52$, $\eta_p^2 = 0.55$, 95% CI (0.48, 0.60), and its environmental impact (i.e. the number of lights illuminated by choosing this option): $F_{1,174} = 16.19$, $\eta_p^2 = 0.18$, 95% CI (0.08, 0.27), exerted marked effects on PEBT choice behaviour. These findings replicate earlier work [34] and suggest that participants take into account both these types of actual consequences when choosing between PEBT options. Neither cost: $F_{2.18, 379.49} = 1.01$, $\eta_p^2 = 0.01$, 95% CI (0.00, 0.03), nor impact: $F_{1,174} = 0.94$, $\eta_p^2 = 0.01$, 95% CI (0.00, 0.05), moderated the effect of observability.

### 3.1.5. Response times

Mean PEBT response times (RTs) were faster in the observable (525 ms, s.d. = 237 ms) than in the non-observable condition (622 ms, s.d. = 253 ms): $d = -0.40$, 95% CI ($-0.70$, $-0.10$). In the observable condition, higher RTs were positively related to participants' privacy ratings: $r = 0.33$, 95% CI (0.14, 0.50). Of note, both response latency: $r = 0.23$, 95% CI (0.03, 0.42), and rated privacy: $r = 0.26$, 95% CI (0.06, 0.44), were positively correlated to the overall proportion of participants' bicycle choices in that condition (table 3). Similar results were obtained when using median RTs instead of mean RTs.

## 4. Discussion

In the present study, the presence of an observer did not promote pro-environmental behaviour on a validated laboratory task. Several manipulation and robustness checks indicate that observability was successfully manipulated and that the null finding was unlikely to be owing to arbitrary analytical choices. Our results suggest that the effect of observability on pro-environmental behaviour might be smaller or less general than expected based on prior literature in the field [13,33].

Our study was designed to have 95% power for detecting an observability effect of $d = 0.50$. If the true effect size is smaller than this estimate, we might have failed to detect it owing to a lack of statistical power. Given our sample size, the likelihood to find a group difference of, for example $d = 0.20$, would only be 37%. Effects of this size can still be of theoretical and practical relevance, but they would render the current laboratory procedure impractical. Given a true effect size of $d = 0.20$, thousands of participants would be required to test theoretically meaningful moderators that might attenuate (e.g. watching eyes versus actual observers) or strengthen (e.g. many observers versus one observer) the effect of observability in a between-subjects design. In combination with the comparatively high cost of laboratory data collection, these recruitment demands would probably discourage future work using the current methods. One possibility to mitigate this issue in future laboratory studies might be the use of within-subject manipulations of observability.

Alternatively, heterogeneous findings on the effect of observability on pro-environmental behaviour may challenge the existence of a single underlying true effect. By asking highly general research questions (such as the one regarding the effect of observability on pro-environmental behaviour), we may neglect important facets of complexity. Observability can take many forms and pro-environmental behaviours differ on a number of potentially relevant dimensions [42,43]. In our study, for example, participants were exposed to physically present observers during the entire choice process, whereas in the study by Vesely & Klöckner [33], only the result of participants' decisions was communicated to the

observer. In situations of the latter type, people can make their decision in private and their decision has no consequence for the further course of the experiment. In contrast to such situations, seating participants next to an observer for the entire task might have interfered with participants' privacy in our study. This assumption is corroborated by lower privacy ratings in the observed than in the non-observed condition of our study. There are numerous benefits of privacy, from informational control to the opportunity to manage bodily functions [44,45], which individuals need to forego when in the presence of potential observers. Individuals may thus tend to escape observable situations as created in our study to restore the benefits of privacy. Our exploratory observation of reduced response latencies in the observable condition is consistent with this notion (but may also reflect effects of social facilitation [46]).

Critically, in our study, anti-environmental behaviour was instrumental in the escape from privacy-violating observability. Choosing the more energy-consuming car option reduced the time participants had to spend on the task and thus in the presence of the observer. This characteristic does not disqualify our procedure for the study of observability effects. Many pro-environmental behaviours involve costs in terms of time and in non-private situations, engaging in these behaviours will prolong the state of privacy violation. Recycling in a cafeteria will take more time (spent in the presence of other cafeteria users) than throwing everything in the same bin. Similarly, taking the train will imply spending more time in the presence of others than taking the plane for many destinations. Such situations involve contingencies similar to our laboratory setting, but different from the experimental situation created in other observability studies [e.g. 33]. It is this similarity (or lack thereof) that determines whether results from laboratory observability studies can be generalized to everyday pro-environmental behaviours performed in public versus private.

As a result, we advocate for more specificity in the analysis of observability effects on pro-environmental behaviours. One type of observability (but not another) may promote a particular pro-environmental behaviour in a given situation while leaving other behaviours unaffected. Ascertaining that a situation can be classified as observable and a behaviour as pro-environmental is probably insufficient to predict the occurrence of an effect. Instead, an increased focus on the consequences of a particular pro-environmental behaviour in a particular observation situation may be helpful. For example, our exploratory results indicate that it may be critical to consider whether the pro-environmental behaviour of interest prolongs the observation situation. However, even when it does, other situational factors might mitigate the aversiveness of the observation situation. Time-consuming pro-environmental behaviour might be unaffected or even promoted by observability when targets know their observers, have been rewarded by them in the past, or can interact with them in the observation situation. A comprehensive analysis of all consequences produced by a particular pro-environmental behaviour seems necessary to predict how this behaviour will be affected by a particular type of observability.

An increased focus on situation- and behaviour-specific consequences has the potential to inspire future research for a better understanding of observability effects. To this end, validated laboratory procedures eliciting consequential behaviour (such as the PEBT) may prove particularly helpful [43]. For example, future PEBT studies can examine whether observability affects behaviour on the task when anti-environmental behaviour does not facilitate escape from observation (e.g. when only the number of bicycle choices is communicated to observers sitting in another room). Similarly, it can be tested whether replacing the task's time consequences with monetary consequences or running the procedure with strangers versus acquaintances as observers modulates the effect of observability. With regard to observability effects on pro-environmental donations (used as a secondary outcome measure and not found to be affected in the present study), it might be worthwhile to explore if effects depend on whether observers can see the amount donated or have a possibility to reward the donor. In our view, such a systematic study taking into account situational and behavioural specifics would offer the most promising way to approach the complexity of observability effects on prosociality in the environmental domain and beyond.

Ethics. The experimental protocol has received approval from the Social and Societal Ethics Committee (SMEC) at KU Leuven (G-2018 11 1394). Informed consent was obtained from all participants.

Data accessibility. All data, study materials and analysis scripts as well as the final version of the Stage 1 protocol are available at https://osf.io/z4jwd/.

Authors' contributions. All authors were involved in the conception and design of the study. F.L. collected and analysed the data. F.L. drafted the manuscript and C.B. and S.D. provided critical revisions. All authors approved the final version of the manuscript.

Competing interests. We declare we have no competing interests.

Funding. F.L. received funding from the FWO and European Union's Horizon 2020 research and innovation programme under the Marie Skłodowska-Curie grant agreement no. 665501.

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
