## [Reviewer comments · Royal Society Open Science]

Review History

RSOS-190189.R0 (Original submission)

Review form: Reviewer 1

Is the language acceptable?

Yes

Do you have any ethical concerns with this paper?

No

Have you any concerns about statistical analyses in this paper?

No

Recommendation?

Major revision

Comments to the Author(s)

While the following review points out a number of opportunities for further development of the design and its presentation, I would like to stress I see much value in the proposed research and I believe it should be possible for the authors to address all concerns raised here.

Ad: 1. The significance of the research question(s)

Decision observability, the central focus of the proposed investigation, is recognized as a particularly important factor in the domain of pro-social and, more recently, pro-environmental behavior research (for

specific comments on the hypotheses see the next section). A highly commendable aspect of the authors' design is the use of a psychometrically validated and incentive-compatible task. Also, it will be useful to have evidence obtained in a pre-registered study.

Ad: 2. The logic, rationale, and plausibility of the proposed hypotheses

The main hypothesis is formulated clearly. The secondary hypotheses, while conceptually clear, could be described in more detail, with reference to variable names, the concrete operationalizations of which are to be provided elsewhere in the text (say, in the Procedures section).

Testing the main hypothesis can be regarded as a conceptual replication of previous research. The novelty of the paper might thus benefit from, in addition, testing one (or more) of what is now framed as exploratory hypotheses in pre-registered a priori tests, alongside the main hypothesis. I am not saying this is necessary, but it seems like an option that should be given some thought.

The scope of the proposed secondary hypotheses seems perhaps unnecessarily broad (see the next section for discussion of some other issues concerning some of these analyses). To me a paper with fewer secondary analyses and hence a more streamlined story would be just as interesting and quite possibly a nicer read. Remember that you will, for example, probably need to touch upon all the additional variables involved in the secondary tests in the exposition of the theory (especially if you actually obtain interesting results)...

As a side note, I would recommend being somewhat more cautious in certain claims the authors make. For example, environmental behavior and people who perform it are not always perceived more positively (p. 4, lines 17-20; see e.g. Welte & Anastasio, 2010 in *Environment and Behavior*; Berger, 2017 in *PloS ONE*). Social norm interventions are not always effective or their effects may be small (p. 4, lines 26-29; see e.g. Allcott, 2011 in *Journal of Public Economics*). Likewise, the criticism of the subtle cues of observation approach would need to be outlined more carefully (p. 4, lines 45-47).

Ad: 3. The soundness and feasibility of the methodology and analysis pipeline (including statistical power analysis where applicable)

Please indicate whether the power calculations are for a one-sided or a two-sided test (p. 7, lines 30-36). Related to this, a two-sided test of the main hypothesis is proposed on p. 14 (lines 58-59), although the directional main hypothesis would seem to allow applying a one-sided test. Or are the authors unsure about the direction of the effect? In that case, this should be mentioned when presenting the hypothesis itself (p. 6, lines 31-38).

In case you subsequently decide to test more than one a priori hypothesis, please indicate which, if any, corrections for testing multiple hypotheses will be employed.

If possible, I would recommend ensuring the following in the observable treatment: (a) the target knows he or she is being observed, (b) the target cannot observe the observer.

Condition (b) does not seem to be satisfied in the current version of the design. To implement it, the observer can be seated 1-2 meters behind the target, for example, and observe the lights going on and off from there (the use of video could be another solution; displaying what the target sees on his or her screen on the observer's screen, while physically isolating the two individuals, could be yet another approach). Importantly, note that unless condition (b) is satisfied, it is impossible to distinguish the effect (on the target's behavior) of being observed from the effect of any number of possible subtle cues intentionally or unintentionally displayed by the observer, such as signs of approval or disapproval, interest or boredom, tranquility or impatience, etc. If you decide to modify the observable treatment, please make sure to modify the non-observable treatment in a corresponding manner, so that the treatments are still comparable (e.g. with respect to where participants are seated).

Concerning condition (a) above, it seems likely that in the current design some observers will not in fact engage in observation and that some targets will assume that the observers are not engaging in observation (see esp. p. 12, lines 51-59 and p. 13, lines 3-8). I realize the authors include manipulation checks that partly alleviate this concern (p. 12, lines 47-49; p. 13, lines 31-34). Still, a design where observers are explicitly tasked with observing the behavior of the targets and the targets are explicitly made aware of this would

seem preferable. I am open to discussing this issue with the authors further, since I do not really understand the reasons behind the authors' design choice here.

An additional concern is the hypothesized treatment effect on some of the secondary outcome measures, most notably the targets' gifts to environmental organizations and the targets' gifts to their respective observers (p. 15, lines 35-40).

With respect to gifts to observers, I am assuming the observer will know that the gift (which he or she will see) comes from the target sitting next to them. It seems to me that this will, critically, be the case to almost the same extent in both the observable and in the non-observable treatment (see Fig. 1 on p. 14). Thus, to postulate a treatment effect here does not seem reasonable.

With respect to gifts to environmental organizations, I am assuming the donation decision will be a one-shot decision, which will be inherently very hard to observe for observers in the observable treatment (they would have to literally stare at their target's screen at the very moment the donation is made). Thus, again, to postulate a treatment effect here does not seem very reasonable.

For the same reason, one should not expect meaningful treatment differences to occur with respect to the observers' estimates of the donations made by their corresponding targets, at least assuming participants' rationality (p. 15, lines 42-45).

Perhaps some effects could occur due to biased cognition (e.g., targets might erroneously believe their donations to environmental organizations are being observed in the observable treatment, even though it seems unlikely that observing the donation would be feasible in the current design). But if the authors want to test for this type of biases, I feel that a rather specific underlying theory would need to be presented. Without having such a theory beforehand, running these additional tests will probably not benefit the paper.

Thus, the authors may want to drop these additional outcome measures and only focus on the main dependent variable. An alternative option would be to modify the procedures in such a way that treatment differences could be realistically expected to occur in case of the additional outcome measures as well (e.g., by explicitly revealing to the observer the target's donation to the environmental organization in the observable treatment, see e.g. Vesely & Klöckner, 2018 in *Journal of Environmental Psychology*).

I also recommend that the authors consider whether meaningful treatment differences can be expected to occur in the correlational analyses outlined on p. 15 (lines 46-59). Please keep only those tests that make sense given the issues just discussed.

Although this is not crucial, if environmental identity is to be used as a moderator of the effect of observability (p. 6, lines 54-59), one should consider placing the environmental identity items before the PEBT task (perhaps inserting a filler task between the two), since performance of pro-environmental behavior can have short-term effects on one's environmental identity perceptions (see e.g. van der Werff et al., 2014 in *Journal of Environmental Psychology*).

Ad: 4. Whether the clarity and degree of methodological detail would be sufficient to replicate exactly the proposed experimental procedures and analysis pipeline
I find the amount of provided detail to be adequate.

Ad: 5. Whether the authors provide a sufficiently clear and detailed description of the methods to prevent undisclosed flexibility in the experimental procedures or analysis pipeline
I do not see any problems here.

Ad: 6. Whether the authors have considered sufficient outcome-neutral conditions (e.g. positive controls) for ensuring that the results obtained are able to test the stated hypotheses
I do not see any problems here (see p. 15., lines 10-17).

Review form: Reviewer 2

Is the language acceptable?

Yes

Do you have any ethical concerns with this paper?

No

Have you any concerns about statistical analyses in this paper?

No

Recommendation?

Accept with minor revision

Comments to the Author(s)

I like the idea for this study - it will be a useful contribution however the results come out.

Well done to the authors for creating a registered report. It is well executed. I have some design comments, and then some more minor wording ones.

Design

It's a great design for a study, but there are one or two issues to think about. Running multiple pairs of participants at the same time in one room is elegant and efficient, but it does also mean that even in the non-observable condition, there are quite a few other people around (there is the non-observing pair partner, and also six other people in the room). So in the event of a null result, the authors may end up wondering if the T players in the non-observed condition still felt a bit observed. I would like the authors to give some rationale as to why they did not run the (to me more obvious) condition of having no-one in the room at all. At the very least I would expect to see some kind of manipulation check that participants felt less observed in the non-observed condition, and that the O players did actually see less. You could ask these questions afterwards. I was also worried that light could flare up in such a way as to still be observable in the non-observed condition (or the T players could fear that it might). In other words, some validation is needed that people really do perceive themselves as less observable in the 'non-observable' condition. Could you provide some pilot data on this?

I was also not sure from the presentation how explicit it is to the participants that they will be able to finish sooner if they take the car. This is critical. For example, if Ps believed that by finishing sooner, they would just end up sitting doing nothing until the last person in the room finished the study, then they would not feel there was any real benefit to taking the car anyway, and the validity might be undermined. How will you manage people finishing at different times?

Page 3, Line 42. Sorry to be a bore, but the Homo economicus model does not require narrow self-interest. It merely requires that individuals consistently choose those options that maximise the value of their utility function. However, it is open on what the different arguments in that function are. Although many Homo economicus models assume income maximisation is what gives people utility, it is not essential to the approach to make this assumption. For example, there are 'social preference' models in which people are assumed to derive utility from the welfare of others, and these are still Homo economicus models. Best to just say 'the assumption of narrow self-interest is sufficient...'

Page 4, line 3. What is meant by 'evolved' here? Not Darwinian evolution presumably.

Page 4, line 40. What I got from the cited review on observability [25] was not that prior findings on observability were mixed, but rather than effects were quite weak (explaining why sometimes they are statistically significant and sometimes not). Also there are various moderators. That's not the same as saying 'mixed' which implies unexplained and large heterogeneity in findings.

Page 5, line 8. I would not say that there is a lack of conclusive evidence for observability effects; on the contrary, it's a large literature and when you meta-analyse it there is something there. Rather, I would agree that the effects people have found are typically quite weak. The limitations you mention could well explain the weakness.

Page 5, line 54. This sounds great, but can you give some more evidence for the external validity of the PEBT? i.e. does it predict some behaviour outside the lab? The sentence beginning 'Psychometric studies...' did not go on to say what these studies did or what they found.

Page 14, Fig 1. I like the design, but I guess whether it works depends on the exact layout of the room and the furniture (T has to feel anonymous in the non-observable condition; O has to really see). Could you include some manipulation check questions about whether they felt watched, and whether O saw? Also, in the procedure, it was not clear enough to me that T knew they could get away sooner if they always took the car. If they thought that by taking the car they would just have to sit around doing nothing at the end of the session, then there would be no personal benefit to doing so. Please clarify that it is the case, and T knows it to be the case, that taking the car actually shortens the session.

Also, is O's music task a fixed duration? Or does O leave when T does? And finally, will you screen out pairs where T and O know each other?

Page 15. In the secondary analyses, I was unclear whether the ANOVAs would also include interactions between observability and the additional variables. This would make sense to me – for example there could well be an interaction between observability and, for example, cost or benefit.

As a final comment, if this study is illuminating I would love to see it replicated with a third condition, artificial eye cues – my prediction is that these would have some effect, but less than the real O. That would be a nice one to test.

Review form: Reviewer 3

Is the language acceptable?

Yes

Do you have any ethical concerns with this paper?

No

Have you any concerns about statistical analyses in this paper?

Yes

Recommendation?

Accept with minor revision

Comments to the Author(s)

With enormous interest, I read the registered report "Green when seen", submitted to RSOS. My background from Social and Environmental Psychology fits to the ideas and predictions developed in this manuscript, and I am happy to provide some feedback.

Overall, I believe that this study may become an important and visible contribution to the literature on pro-environmental behavior. I know previous work of the authors using the PEBT task, and I find it an intriguing and useful method to come as close as possible to something like "real behavior" in a lab setting. Also, the manuscript is elegantly written, considering both recent and classic relevant literature.

There are some suggestions I would like to share that I believe could improve the manuscript and the study.

1) In line 28/29 on p.3 the authors claim that "our knowledge is still limited about the factors that give rise to pro-environmental behaviour." This is somewhat trivial as this applies to virtually every scientific knowledge. I do not believe there is need for such a statement, I think stating that environmental psychology research could use more behavioral measures would be a sufficient and relevant point to make.

2) p3, l54ff: I am not yet fully convinced of the theoretical asrgument the authors try to make. While I applaud to link "observability" to social norms, I am not sure that the same processes act here. The authors suggest that norm-compliers being more attractive, respected, sophisticated and the such is directly linked to pro-environmental compliance under observability. Is this a logical consequence? Not for me. This needs either more lines to fully make this link clear, or some other process explanation (see 3).

3) Have the authors considered "accountability" as an alternative explanation? Accountability and "being observed" are strongly intertwined and often confounded. Being held accountable for one's actions may be a particularly strong driver of pro-environmental behavior (which can be easily observed when discussing with fellow scientists about "flying").

4) Will the pairs of observers and targets be matched, e.g. in terms of gender/age? Not because I believe these are relevant psychological constructs here, but simply because it could make a difference whether I am accountable to an in-group or an outgroup member. Also, we know from pain research that men seem to accept more/longer pain when females are around - similarly, they could prove to be more "environmental" when an other-sex rather than same-sex member is around.

5) The secondary hypotheses come a bit sudden - what is the rationale for using these (cost, impact,...) and not others? I understand these are exploratory but there should be some suggestion why these should be particularly relevant.

6) Finally, for the analysis, I would suggest to think over the power analysis again. Yes, for a simple main effect the power analysis is correct, but I believe the more tests the authors make, and the more factors and covariates are involved, the less meaningful and appropriate the power analysis becomes. I do not believe that "we" need extraordinarily large samples (off the protocol: this is only one of various issues psychological research needs to address, besides better theories) but to make your research coherent, I would suggest to elaborate a little bit more here.

All in all, I believe this work can make an excellent contribution, and I hope the authors perceive my suggestions as what they are: Suggestions, not criticism of their work. I wish the authors all the best with this research line.

Decision letter (RSOS-190189.R0)

25-Feb-2019

Dear Dr Lange,

The Editors assigned to your Stage 1 Registered Report ("Green when seen? Testing the effect of observability on prosociality in an environmental conservation task") have now received comments from reviewers. We would like you to revise your paper in accordance with the referee and editors suggestions which can be found below (not including confidential reports to the Editor). Please note this decision does not guarantee eventual acceptance.

When submitting your revised manuscript, you must respond to the comments made by the referees and upload a file "Response to Referees" in "Section 2 - File Upload". Please use this to document how you have responded to the comments, and the adjustments you have made. In order to expedite the processing of the revised manuscript, please be as specific as possible in your response.

Kind regards,
Professor Chris Chambers
Royal Society Open Science
openscience@royalsociety.org

on behalf of Professor Chris Chambers (Registered Reports Editor, Royal Society Open Science)
 openscience@royalsociety.org

Associate Editor Comments to Author (Professor Chris Chambers):

Three expert reviewers have now appraised the manuscript. All find merit in the submission but also point out several areas that would benefit from improvement, including theory and rationale, suitability of control conditions (including inclusion of an appropriate manipulation check), and precision of the hypotheses. A major revision is therefore recommended. In revising the manuscript, please pay particular attention to two issues: ensuring that the hypotheses are specified precisely in terms of specific independent and dependent variables, and that there is a clear and direct mapping between each hypothesis and the corresponding statistical test (or test component) that will confirm or disconfirm that hypothesis. One straightforward way to do this is to present the hypotheses in a numbered list style, and then the analyses in a corresponding list style (e.g. in the text or in a table). For the secondary hypotheses, especially, these links are not yet sufficiently clear. Please also ensure that the power analyses correspond directly these tests -- in other words, every hypothesis test should be accompanied by a corresponding power analysis, whereas at present the power analysis is based entirely on just one of the tests -- an independent t-test.

Comments to Author:

Reviewer: 1

Comments to the Author(s)

While the following review points out a number of opportunities for further development of the design and its presentation, I would like to stress I see much value in the proposed research and I believe it should be possible for the authors to address all concerns raised here.

Ad: 1. The significance of the research question(s)

Decision observability, the central focus of the proposed investigation, is recognized as a particularly important factor in the domain of pro-social and, more recently, pro-environmental behavior research (for specific comments on the hypotheses see the next section). A highly commendable aspect of the authors' design is the use of a psychometrically validated and incentive-compatible task. Also, it will be useful to have evidence obtained in a pre-registered study.

Ad: 2. The logic, rationale, and plausibility of the proposed hypotheses

The main hypothesis is formulated clearly. The secondary hypotheses, while conceptually clear, could be described in more detail, with reference to variable names, the concrete operationalizations of which are to be provided elsewhere in the text (say, in the Procedures section).

Testing the main hypothesis can be regarded as a conceptual replication of previous research. The novelty of the paper might thus benefit from, in addition, testing one (or more) of what is now framed as exploratory hypotheses in pre-registered a priori tests, alongside the main hypothesis. I am not saying this is necessary, but it seems like an option that should be given some thought.

The scope of the proposed secondary hypotheses seems perhaps unnecessarily broad (see the next section for discussion of some other issues concerning some of these analyses). To me a paper with fewer secondary analyses and hence a more streamlined story would be just as interesting and quite possibly a nicer read. Remember that you will, for example, probably need to touch upon all the additional variables involved in the secondary tests in the exposition of the theory (especially if you actually obtain interesting results)...

As a side note, I would recommend being somewhat more cautious in certain claims the authors make. For example, environmental behavior and people who perform it are not always perceived more positively (p. 4, lines 17-20; see e.g. Welte & Anastasio, 2010 in *Environment and Behavior*; Berger, 2017 in *PLoS ONE*). Social norm interventions are not always effective or their effects may be small (p. 4, lines 26-29; see e.g. Allcott, 2011 in *Journal of Public Economics*). Likewise, the criticism of the subtle cues of observation approach would need to be outlined more carefully (p. 4, lines 45-47).

Ad: 3. The soundness and feasibility of the methodology and analysis pipeline (including statistical power analysis where applicable)
Please indicate whether the power calculations are for a one-sided or a two-sided test (p. 7, lines 30-36). Related to this, a two-sided test of the main hypothesis is proposed on p. 14 (lines 58-59), although the directional main hypothesis would seem to allow applying a one-sided test. Or are the authors unsure about the direction of the effect? In that case, this should be mentioned when presenting the hypothesis itself (p. 6, lines 31-38).

In case you subsequently decide to test more than one a priori hypothesis, please indicate which, if any, corrections for testing multiple hypotheses will be employed.

If possible, I would recommend ensuring the following in the observable treatment: (a) the target knows he or she is being observed, (b) the target cannot observe the observer.

Condition (b) does not seem to be satisfied in the current version of the design. To implement it, the observer can be seated 1-2 meters behind the target, for example, and observe the lights going on and off from there (the use of video could be another solution; displaying what the target sees on his or her screen on the observer's screen, while physically isolating the two individuals, could be yet another approach). Importantly, note that unless condition (b) is satisfied, it is impossible to distinguish the effect (on the target's behavior) of being observed from the effect of any number of possible subtle cues intentionally or unintentionally displayed by the observer, such as signs of approval or disapproval, interest or boredom, tranquility or impatience, etc. If you decide to modify the observable treatment, please make sure to modify the non-observable treatment in a corresponding manner, so that the treatments are still comparable (e.g. with respect to where participants are seated).

Concerning condition (a) above, it seems likely that in the current design some observers will not in fact engage in observation and that some targets will assume that the observers are not engaging in observation (see esp. p. 12, lines 51-59 and p. 13, lines 3-8). I realize the authors include manipulation checks that partly alleviate this concern (p. 12, lines 47-49; p. 13, lines 31-34). Still, a design where observers are explicitly tasked with observing the behavior of the targets and the targets are explicitly made aware of this would seem preferable. I am open to discussing this issue with the authors further, since I do not really understand the reasons behind the authors' design choice here.

An additional concern is the hypothesized treatment effect on some of the secondary outcome measures, most notably the targets' gifts to environmental organizations and the targets' gifts to their respective observers (p. 15, lines 35-40).

With respect to gifts to observers, I am assuming the observer will know that the gift (which he or she will see) comes from the target sitting next to them. It seems to me that this will, critically, be the case to almost the same extent in both the observable and in the non-observable treatment (see Fig. 1 on p. 14). Thus, to postulate a treatment effect here does not seem reasonable.

With respect to gifts to environmental organizations, I am assuming the donation decision will be a one-shot decision, which will be inherently very hard to observe for observers in the observable treatment (they would have to literally stare at their target's screen at the very moment the donation is made). Thus, again, to postulate a treatment effect here does not seem very reasonable.

For the same reason, one should not expect meaningful treatment differences to occur with respect to the observers' estimates of the donations made by their corresponding targets, at least assuming participants' rationality (p. 15, lines 42-45).

Perhaps some effects could occur due to biased cognition (e.g., targets might erroneously believe their donations to environmental organizations are being observed in the observable treatment, even though it seems unlikely that observing the donation would be feasible in the current design). But if the authors want

to test for this type of biases, I feel that a rather specific underlying theory would need to be presented. Without having such a theory beforehand, running these additional tests will probably not benefit the paper.

Thus, the authors may want to drop these additional outcome measures and only focus on the main dependent variable. An alternative option would be to modify the procedures in such a way that treatment differences could be realistically expected to occur in case of the additional outcome measures as well (e.g., by explicitly revealing to the observer the target's donation to the environmental organization in the observable treatment, see e.g. Vesely & Klöckner, 2018 in *Journal of Environmental Psychology*).

I also recommend that the authors consider whether meaningful treatment differences can be expected to occur in the correlational analyses outlined on p. 15 (lines 46-59). Please keep only those tests that make sense given the issues just discussed.

Although this is not crucial, if environmental identity is to be used as a moderator of the effect of observability (p. 6, lines 54-59), one should consider placing the environmental identity items before the PEBT task (perhaps inserting a filler task between the two), since performance of pro-environmental behavior can have short-term effects on one's environmental identity perceptions (see e.g. van der Werff et al., 2014 in *Journal of Environmental Psychology*).

Ad: 4. Whether the clarity and degree of methodological detail would be sufficient to replicate exactly the proposed experimental procedures and analysis pipeline
I find the amount of provided detail to be adequate.

Ad: 5. Whether the authors provide a sufficiently clear and detailed description of the methods to prevent undisclosed flexibility in the experimental procedures or analysis pipeline
I do not see any problems here.

Ad: 6. Whether the authors have considered sufficient outcome-neutral conditions (e.g. positive controls) for ensuring that the results obtained are able to test the stated hypotheses
I do not see any problems here (see p. 15., lines 10-17).

Reviewer: 2

Comments to the Author(s)

I like the idea for this study - it will be a useful contribution however the results come out.

Well done to the authors for creating a registered report. It is well executed. I have some design comments, and then some more minor wording ones.

Design

It's a great design for a study, but there are one or two issues to think about. Running multiple pairs of participants at the same time in one room is elegant and efficient, but it does also mean that even in the non-observable condition, there are quite a few other people around (there is the non-observing pair partner, and also six other people in the room). So in the event of a null result, the authors may end up wondering if the T players in the non-observed condition still felt a bit observed. I would like the authors to give some rationale as to why they did not run the (to me more obvious) condition of having no-one in the room at all. At the very least I would expect to see some kind of manipulation check that participants felt less observed in the non-observed condition, and that the O players did actually see less. You could ask these questions afterwards. I was also worried that light could flare up in such a way as to still be observable in the non-observed condition (or the T players could fear that it might). In other words, some validation is needed that people really do perceive themselves as less observable in the 'non-observable' condition. Could you provide some pilot data on this?

I was also not sure from the presentation how explicit it is to the participants that they will be able to finish sooner if they take the car. This is critical. For example, if Ps believed that by finishing sooner, they would just end up sitting doing nothing until the last person in the room finished the study, then they would not

feel there was any real benefit to taking the car anyway, and the validity might be undermined. How will you manage people finishing at different times?

Page 3, Line 42. Sorry to be a bore, but the Homo economicus model does not require narrow self-interest. It merely requires that individuals consistently choose those options that maximise the value of their utility function. However, it is open on what the different arguments in that function are. Although many Homo economicus models assume income maximisation is what gives people utility, it is not essential to the approach to make this assumption. For example, there are 'social preference' models in which people are assumed to derive utility from the welfare of others, and these are still Homo economicus models. Best to just say 'the assumption of narrow self-interest is sufficient...'

Page 4, line 3. What is meant by 'evolved' here? Not Darwinian evolution presumably.

Page 4, line 40. What I got from the cited review on observability [25] was not that prior findings on observability were mixed, but rather than effects were quite weak (explaining why sometimes they are statistically significant and sometimes not). Also there are various moderators. That's not the same as saying 'mixed' which implies unexplained and large heterogeneity in findings.

Page 5, line 8. I would not say that there is a lack of conclusive evidence for observability effects; on the contrary, it's a large literature and when you meta-analyse it there is something there. Rather, I would agree that the effects people have found are typically quite weak. The limitations you mention could well explain the weakness.

Page 5, line 54. This sounds great, but can you give some more evidence for the external validity of the PEBT? i.e. does it predict some behaviour outside the lab? The sentence beginning 'Psychometric studies...' did not go on to say what these studies did or what they found.

Page 14, Fig 1. I like the design, but I guess whether it works depends on the exact layout of the room and the furniture (T has to feel anonymous in the non-observable condition; O has to really see). Could you include some manipulation check questions about whether they felt watched, and whether O saw? Also, in the procedure, it was not clear enough to me that T knew they could get away sooner if they always took the car. If they thought that by taking the car they would just have to sit around doing nothing at the end of the session, then there would be no personal benefit to doing so. Please clarify that it is the case, and T knows it to be the case, that taking the car actually shortens the session.

Also, is O's music task a fixed duration? Or does O leave when T does? And finally, will you screen out pairs where T and O know each other?

Page 15. In the secondary analyses, I was unclear whether the ANOVAs would also include interactions between observability and the additional variables. This would make sense to me - for example there could well be an interaction between observability and, for example, cost or benefit.

As a final comment, if this study is illuminating I would love to see it replicated with a third condition, artificial eye cues - my prediction is that these would have some effect, but less than the real O. That would be a nice one to test.

Reviewer: 3

Comments to the Author(s)

With enormous interest, I read the registered report "Green when seen", submitted to RSOS. My background from Social and Environmental Psychology fits to the ideas and predictions developed in this manuscript, and I am happy to provide some feedback.

Overall, I believe that this study may become an important and visible contribution to the literature on pro-environmental behavior. I know previous work of the authors using the PEBT task, and I find it an intriguing and useful method to come as close as possible to something like "real behavior" in a lab setting. Also, the manuscript is elegantly written, considering both recent and classic relevant literature.

There are some suggestions I would like to share that I believe could improve the manuscript and the study.

1) In line 28/29 on p.3 the authors claim that "our knowledge is still limited about the factors that give rise to pro-environmental behaviour." This is somewhat trivial as this applies to virtually every scientific knowledge. I do not believe there is need for such a statement, I think stating that environmental psychology research could use more behavioral measures would be a sufficient and relevant point to make.

2) p3, l54ff: I am not yet fully convinced of the theoretical argument the authors try to make. While I

applaud to link "observability" to social norms, I am not sure that the same processes act here. The authors suggest that norm-compliers being more attractive, respected, sophisticated and the such is directly linked to pro-environmental compliance under observability. Is this a logical consequence? Not for me. This needs either more lines to fully make this link clear, or some other process explanation (see 3).

3) Have the authors considered "accountability" as an alternative explanation? Accountability and "being observed" are strongly intertwined and often confounded. Being held accountable for one's actions may be a particularly strong driver of pro-environmental behavior (which can be easily observed when discussing with fellow scientists about "flying").

4) Will the pairs of observers and targets be matched, e.g. in terms of gender/ age? Not because I believe these are relevant psychological constructs here, but simply because it could make a difference whether I am accountable to an in-group or an outgroup member. Also, we know from pain research that men seem to accept more/longer pain when females are around - similarly, they could prove to be more "environmental" when an other-sex rather than same-sex member is around.

5) The secondary hypotheses come a bit sudden - what is the rationale for using these (cost, impact,...) and not others? I understand these are exploratory but there should be some suggestion why these should be particularly relevant.

6) Finally, for the analysis, I would suggest to think over the power analysis again. Yes, for a simple main effect the power analysis is correct, but I believe the more tests the authors make, and the more factors and covariates are involved, the less meaningful and appropriate the power analysis becomes. I do not believe that "we" need extraordinarily large samples (off the protocol: this is only one of various issues psychological research needs to address, besides better theories) but to make your research coherent, I would suggest to elaborate a little bit more here.

All in all, I believe this work can make an excellent contribution, and I hope the authors perceive my suggestions as what they are: Suggestions, not criticism of their work. I wish the authors all the best with this research line.

Author's Response to Decision Letter for (RSOS-190189.R0)

See Appendix A.

RSOS-190189.R1 (Revision)

Review form: Reviewer 1

Is the language acceptable?

Yes

Do you have any ethical concerns with this paper?

No

Have you any concerns about statistical analyses in this paper?

No

Recommendation?

Accept in principle

Comments to the Author(s)

I am happy with how the authors addressed my comments on the previous draft and I recommend the prospective study to be accepted for publication in this journal. The only remaining potential issue I see is the fact that the authors decided to decrease the sample size for their study. Yes, the sample size is adequate for the one-sided test, but in my opinion it would strengthen the study if the authors could be slightly more conservative in the power calculations. Only a handful of related prior studies (pp. 8-9, authors' pagination) were considered when estimating the likely effect size for the planned experiment, which causes some uncertainty as to what effect size to expect (given the possible existence of a few unpublished null results).

Review form: Reviewer 2**Is the language acceptable?**

Yes

Do you have any ethical concerns with this paper?

No

Have you any concerns about statistical analyses in this paper?

No

Recommendation?

Accept in principle

Comments to the Author(s)

Thanks to the authors for making detailed changes in response to comments. I am satisfied that they responded to what I raised. This has been a nice, constructive process and I wish the authors all the best as they now complete their study.

Review form: Reviewer 3**Is the language acceptable?**

Yes

Do you have any ethical concerns with this paper?

No

Have you any concerns about statistical analyses in this paper?

No

Recommendation?

Accept in principle

Comments to the Author(s)

I have now read the revision of the registered report "Green when seen? Testing the effect of observability on prosociality in an environmental conservation task: a Registered Report" and based on my previous review, I find that the authors did very well in strengthening both the rationale and the methodological aspects of the paper. I wish the authors all the best for their study, and look forward to learning about its results.

Decision letter (RSOS-190189.R1)

19-Jun-2019

Dear Dr Lange

On behalf of the Editor, I am pleased to inform you that your Stage 1 Registered Report RSOS-190189.R1 entitled "Green when seen? Testing the effect of observability on prosociality in an environmental conservation task: a Registered Report" has been accepted in principle for publication in Royal Society Open Science. The reviewers' and editors' comments are included at the end of this email.

You may now progress to Stage 2 and complete the study as approved. Before commencing data collection we ask that you:

- 1) Update the journal office as to the anticipated completion date of your study.
- 2) Register your approved protocol on the Open Science Framework (<https://osf.io/rr>) or other recognised repository, either publicly or privately under embargo until submission of the Stage 2 manuscript. Please note that a time-stamped, independent registration of the protocol is mandatory under journal policy, and manuscripts that do not conform to this requirement cannot be considered at Stage 2. The protocol should be registered unchanged from its current approved state, with the time-stamp preceding implementation of the approved study design. **Please note, however, the Associate Editor's comment below which may lead you to slightly revise the protocol if you wish**

Following completion of your study, we invite you to resubmit your paper for peer review as a Stage 2 Registered Report. Please note that your manuscript can still be rejected for publication at Stage 2 if the Editors consider any of the following conditions to be met:

- The results were unable to test the authors' proposed hypotheses by failing to meet the approved outcome-neutral criteria.
- The authors altered the Introduction, rationale, or hypotheses, as approved in the Stage 1 submission.
- The authors failed to adhere closely to the registered experimental procedures. Please note that any deviations from the approved experimental procedures must be communicated to the editor immediately for approval, and prior to the completion of data collection. Failure to do so can result in revocation of in-principle acceptance and rejection at Stage 2 (see complete guidelines for further information).
- Any post-hoc (unregistered) analyses were either unjustified, insufficiently caveated, or overly dominant in shaping the authors' conclusions.
- The authors' conclusions were not justified given the data obtained.

We encourage you to read the complete guidelines for authors concerning Stage 2 submissions at <http://rsos.royalsocietypublishing.org/content/registered-reports>. Please especially note the requirements for data sharing, reporting the URL of the independently registered protocol, and that withdrawing your manuscript will result in publication of a Withdrawn Registration.

Once again, thank you for submitting your manuscript to Royal Society Open Science and we look forward to receiving your Stage 2 submission. If you have any questions at all, please do not hesitate to get in touch. We look forward to hearing from you shortly with the anticipated submission date for your stage two manuscript.

Associate Editor Comments to Author (Professor Chris Chambers):

All reviewers are now satisfied and recommend Stage 1 in principle acceptance. Reviewer 1 notes some

slight dissatisfaction with the reduction in sample size. Having re-read the author's sampling plan, I feel it is sound and sufficient for acceptance at RSOS, however the reviewer does make a valid point and a larger sample size would be preferable. I will therefore leave this to the judgment of the authors. If you wish to proceed with the current sampling plan, then please register the manuscript as it is now, as instructed above at <http://osf.io/rr/>. However, if you wish to increase the sample size slightly to address this point then please let the journal office know and we will reopen the submission portal so that you can submit a further minor revision (which would be assessed rapidly by the editor only).

Reviewers' comments to Author:

Reviewer: 1

Comments to the Author(s)

I am happy with how the authors addressed my comments on the previous draft and I recommend the prospective study to be accepted for publication in this journal. The only remaining potential issue I see is the fact that the authors decided to decrease the sample size for their study. Yes, the sample size is adequate for the one-sided test, but in my opinion it would strengthen the study if the authors could be slightly more conservative in the power calculations. Only a handful of related prior studies (pp. 8-9, authors' pagination) were considered when estimating the likely effect size for the planned experiment, which causes some uncertainty as to what effect size to expect (given the possible existence of a few unpublished null results).

Reviewer: 2

Comments to the Author(s)

Thanks to the authors for making detailed changes in response to comments. I am satisfied that they responded to what I raised. This has been a nice, constructive process and I wish the authors all the best as they now complete their study.

Reviewer: 3

Comments to the Author(s)

I have now read the revision of the registered report "Green when seen? Testing the effect of observability on prosociality in an environmental conservation task: a Registered Report" and based on my previous review, I find that the authors did very well in strengthening both the rationale and the methodological aspects of the paper. I wish the authors all the best for their study, and look forward to learning about its results.

Author's Response to Decision Letter for (RSOS-190189.R1)

See Appendix B.

RSOS-190189.R2 (Revision)

Review form: Reviewer 1

Is the manuscript scientifically sound in its present form?

Yes

Are the interpretations and conclusions justified by the results?

Yes

Is the language acceptable?

Yes

Do you have any ethical concerns with this paper?

No

Have you any concerns about statistical analyses in this paper?

No

Recommendation?

Accept with minor revision

Comments to the Author(s)

Below I briefly address those points that are a standard part of the review criteria for this Journal, and point out a couple of smaller issues. I have no major criticisms. I was very pleased especially with the thought-provoking discussion of the results. When noting in the Discussion that a “validated” laboratory task was used (p. 21, line 47; authors’ pagination used throughout this report), the authors could perhaps further underscore their (exploratory) finding of choice alternatives’ costliness and environmental impact both being associated with choices, while not moderating the effect of observability on choices (p. 20, lines 26-36). This seems to support the interpretation that the absence of an effect of observability was NOT due to participants simply perceiving behavior in the PEBT task as environmentally, personally (and by extension, morally) trivial.

1) Are the data able to test the authors’ proposed hypotheses by passing the approved outcome-neutral criteria (such as absence of floor and ceiling effects or success of positive controls)?

Yes.

2) Are the Introduction, rationale and stated hypotheses the same as the approved Stage 1 submission?

Yes.

3) Did the authors adhere precisely to the registered experimental procedures?

Yes.

4) Are all unregistered exploratory statistical analyses justified, methodologically sound, and informative?

Yes. However, some of these results are left without any comments from the authors (for example “Social consequences of pro-environmental behaviour”, see pp. 19-20).

5) Are the authors’ conclusions justified given the data?

Yes. The issue of relatively limited power is properly discussed (e.g. on p. 22). The authors could consider briefly discussing the possibility of conducting studies where observability is manipulated also within-subject to increase power. It could benefit other researchers to know the authors’ take on this issue.

Small issues: British vs. American English spelling is not used consistently (see e.g. p. 19, lines 56-57 vs. p. 20, lines 33-34). Not all sources are referenced consistently in the text and in the reference list (e.g., Vesely & Klöckner, 2018 appears under the number 31 on p. 22, but under the number 33 in the reference list). Some of the exploratory analyses are not at all discussed, which I personally did not mind, but it seemed a bit unorthodox.

Review form: Reviewer 2

Is the manuscript scientifically sound in its present form?

Yes

Are the interpretations and conclusions justified by the results?

Yes

Is the language acceptable?

Yes

Do you have any ethical concerns with this paper?

No

Have you any concerns about statistical analyses in this paper?

No

Recommendation?

Accept as is

Comments to the Author(s)

The nice thing about reviewing a stage two registered report is that almost everything is already sorted, except the data. The authors have done a great job and report the results very well, followed by a thoughtful discussion.

The authors followed their planned protocol, achieved their sample size, and analyse their data as planned. They present appropriate exploratory analyses and the discussion and inferences are appropriate.

Only some tiny comments:

- Will the paper show somewhere that this was a registered report (e.g. in the title or heading or intro?) I think this is a real strength and readers should know.
- Page 20, line 35. I take it the F-ratios referred to here are for the interaction. I would prefer to see them all given in full with their degrees of freedom.
- Page 20, line 53. I was a bit surprised by this mediation analysis, since in a sense there is nothing to mediate. But it is fine to report it. However for what it is worth there is a significant indirect effect, in the absence of any direct one. What does this mean? Is this worth discussing more in the discussion, and perhaps explaining a bit more here in the results.

Review form: Reviewer 3

Is the manuscript scientifically sound in its present form?

Yes

Are the interpretations and conclusions justified by the results?

Yes

Is the language acceptable?

Yes

Do you have any ethical concerns with this paper?

No

Have you any concerns about statistical analyses in this paper?

No

Recommendation?

Accept as is

Comments to the Author(s)

I have now had the opportunity to read the full manuscript, now entitled "Green when seen? No support for an effect of observability on prosociality in an environmental conservation Task".

In the previous two rounds, I have reviewed and evaluated the manuscript and found that the authors were very sensitive to the reviewers' comments (not just to mine). I applaud the authors for now presenting a stringent Analysis of the actual Experiment. The finding is interesting and useful in communicating that - as usual - it is not "that simple" when it comes to explaining environmental psych phenomena.

At the same time, I find the discussion balanced and critical, allowing room and necessity for further research looking at the boundary conditions of the when-seen effect. Overall, I find this a very sound contribution that convincingly shows the usefulness of pre-registration and succinct conduct of a well-planned protocol.

Decision letter (RSOS-190189.R2)

03-Mar-2020

Dear Dr Lange:

On behalf of the Editor, I am pleased to inform you that your Stage 2 Registered Report RSOS-190189.R2 entitled "Green when seen? No support for an effect of observability on prosociality in an environmental conservation task" has been deemed suitable for publication in Royal Society Open Science subject to minor revision in accordance with the referee suggestions. Please find the referees' comments at the end of this email.

The reviewers and Subject Editor have recommended publication, but also suggest some minor revisions to your manuscript. Therefore, I invite you to respond to the comments and revise your manuscript.

Please also ensure that all the below editorial sections are included where appropriate -- if any section is not applicable to your manuscript, please can we ask you to nevertheless include the heading, but explicitly state that the heading is inapplicable. An example of these sections is attached with this email.

- Ethics statement

- Data accessibility

If you wish to submit your supporting data or code to Dryad (<http://datadryad.org/>), or modify your current submission to dryad, please use the following link:
[http://datadryad.org/submit?journalID=RSOS&manu=\(Document not available\)](http://datadryad.org/submit?journalID=RSOS&manu=(Document not available))

- Competing interests

- Authors' contributions

- Acknowledgements

- Funding statement

Because the schedule for publication is very tight, it is a condition of publication that you submit the revised version of your manuscript within 7 days (i.e. by the 11-Mar-2020). If you do not think you will be able to meet this date please let me know immediately.

Please note that Royal Society Open Science will introduce article processing charges for all new submissions received from 1 January 2018. Registered Reports submitted and accepted after this date will ONLY be subject to a charge if they subsequently progress to and are accepted as Stage 2 Registered Reports. If your manuscript is submitted and accepted for publication after 1 January 2018 (i.e. as a full Stage 2 Registered Report), you will be asked to pay the article processing charge, unless you request a waiver and this is approved by Royal Society Publishing. You can find out more about the charges at <https://royalsocietypublishing.org/rsos/charges>. Should you have any queries, please contact openscience@royalsociety.org.

on behalf of Professor Chris Chambers
(Registered Reports Editor, Royal Society Open Science)
openscience@royalsociety.org

Associate Editor Comments to Author (Professor Chris Chambers):
Comments to the Author:

The Stage 2 manuscript was returned to the three reviewers who assessed it at Stage 1. Happily, all are satisfied with the submission. Reviewer 1 offers some suggestions for minor revision which I would ask the authors to consider before final acceptance is awarded (being careful not to let any additional interpretation of exploratory findings dominate the conclusions).

Comments to Author:
Reviewer: 1

Comments to the Author(s)

Below I briefly address those points that are a standard part of the review criteria for this Journal, and point out a couple of smaller issues. I have no major criticisms. I was very pleased especially with the thought-provoking discussion of the results. When noting in the Discussion that a “validated” laboratory task was used (p. 21, line 47; authors’ pagination used throughout this report), the authors could perhaps further underscore their (exploratory) finding of choice alternatives’ costliness and environmental impact both being associated with choices, while not moderating the effect of observability on choices (p. 20, lines 26-36). This seems to support the interpretation that the absence of an effect of observability was NOT due to participants simply perceiving behavior in the PEBT task as environmentally, personally (and by extension, morally) trivial.

1) Are the data able to test the authors’ proposed hypotheses by passing the approved outcome-neutral criteria (such as absence of floor and ceiling effects or success of positive controls)?

Yes.

2) Are the Introduction, rationale and stated hypotheses the same as the approved Stage 1 submission?

Yes.

3) Did the authors adhere precisely to the registered experimental procedures?

Yes.

4) Are all unregistered exploratory statistical analyses justified, methodologically sound, and informative?

Yes. However, some of these results are left without any comments from the authors (for example “Social consequences of pro-environmental behaviour”, see pp. 19-20).

5) Are the authors’ conclusions justified given the data?

Yes. The issue of relatively limited power is properly discussed (e.g. on p. 22). The authors could consider briefly discussing the possibility of conducting studies where observability is manipulated also within-subject to increase power. It could benefit other researchers to know the authors’ take on this issue.

Small issues: British vs. American English spelling is not used consistently (see e.g. p. 19, lines 56-57 vs. p. 20, lines 33-34). Not all sources are referenced consistently in the text and in the reference list (e.g., Vesely & Klöckner, 2018 appears under the number 31 on p. 22, but under the number 33 in the reference list). Some

of the exploratory analyses are not at all discussed, which I personally did not mind, but it seemed a bit unorthodox.

Reviewer: 2

Comments to the Author(s)

The nice thing about reviewing a stage two registered report is that almost everything is already sorted, except the data. The authors have done a great job and report the results very well, followed by a thoughtful discussion.

The authors followed their planned protocol, achieved their sample size, and analyse their data as planned. They present appropriate exploratory analyses and the discussion and inferences are appropriate.

Only some tiny comments:

- Will the paper show somewhere that this was a registered report (e.g. in the title or heading or intro?) I think this is a real strength and readers should know.
- Page 20, line 35. I take it the F-ratios referred to here are for the interaction. I would prefer to see them all given in full with their degrees of freedom.
- Page 20, line 53. I was a bit surprised by this mediation analysis, since in a sense there is nothing to mediate. But it is fine to report it. However for what it is worth there is a significant indirect effect, in the absence of any direct one. What does this mean? Is this worth discussing more in the discussion, and perhaps explaining a bit more here in the results.

Reviewer: 3

Comments to the Author(s)

I have now had the opportunity to read the full manuscript, now entitled "Green when seen? No support for an effect of observability on prosociality in an environmental conservation Task".

In the previous two rounds, I have reviewed and evaluated the manuscript and found that the authors were very sensitive to the reviewers' comments (not just to mine). I applaud the authors for now presenting a stringent Analysis of the actual Experiment. The finding is interesting and useful in communicating that - as usual - it is not "that simple" when it comes to explaining environmental psych phenomena.

At the same time, I find the discussion balanced and critical, allowing room and necessity for further research looking at the boundary conditions of the when-seen effect.

Overall, I find this a very sound contribution that convincingly shows the usefulness of pre-registration and succinct conduct of a well-planned protocol.

Author's Response to Decision Letter for (RSOS-190189.R2)

See Appendix C.

Decision letter (RSOS-190189.R3)

09-Mar-2020

Dear Dr Lange,

It is a pleasure to accept your manuscript entitled "Green when seen? No support for an effect of observability on environmental conservation in the lab: a Registered Report" in its current form for publication in Royal Society Open Science. Congratulations on an excellent piece of research.

You can expect to receive a proof of your article in the near future. Please contact the editorial office (openscience_proofs@royalsociety.org) and the production office (openscience@royalsociety.org) to let us

know if you are likely to be away from e-mail contact -- if you are going to be away, please nominate a co-author (if available) to manage the proofing process, and ensure they are copied into your email to the journal.

Kind regards,
Lianne Parkhouse
Editorial Coordinator
Royal Society Open Science
openscience@royalsociety.org

on behalf of Professor Chris Chambers (Subject Editor)
openscience@royalsociety.org

Appendix A

Response to the reviewers

We would like to thank the editor and the reviewers for their positive evaluation of our work and the useful suggestions which allowed us to further improve our manuscript. We found the comments to be extraordinarily helpful and believe that our revised submission benefitted a lot. Let us also express our gratitude for the constructiveness of all reviews and the efforts that all parties have invested into making this Registered Report as informative as possible. Below you find a list of the changes that we made in response to the comments raised in the review process.

Editor's comments		
Comment	Response	Change (page numbers refer to the revised version of the manuscript)
Three expert reviewers have now appraised the manuscript. All find merit in the submission but also point out several areas that would benefit from improvement, including theory and rationale, suitability of control conditions (including inclusion of an appropriate manipulation check), and precision of the hypotheses. A major revision is therefore recommended. In revising the manuscript, please pay particular attention to two issues: ensuring that the hypotheses are specified precisely in terms of specific independent and dependent variables, and that there is a clear and direct mapping between each hypothesis and the corresponding statistical test (or test component) that will confirm or disconfirm that hypothesis. One straightforward way to do this is to present the hypotheses in a numbered list style, and then the analyses in a corresponding list style (e.g. in the text or	Thank you very much for facilitating such a constructive peer-review process and for helping us to improve our contribution. In view of your feedback and the reviewers' comments, we have decided to dispense with the secondary hypotheses and to focus entirely on the main hypothesis, for which we can guarantee to have sufficient statistical power. Given the present state of the literature, we consider it more beneficial to establish the main effect of observability on pro-environmental behaviour with improved methodology first. The exploratory data with regard to potential boundary conditions and moderators that we generate in the process can then be used to inform the design of adequately powered follow-up studies.	Deletions on pages 7, 8, & 17

in a table). For the secondary hypotheses, especially, these links are not yet sufficiently clear. Please also ensure that the power analyses correspond directly these tests -- in other words, every hypothesis test should be accompanied by a corresponding power analysis, whereas at present the power analysis is based entirely on just one of the tests -- an independent t-test.		
Reviewer 1		
Comment	Response	Change (page numbers refer to the revised version of the manuscript)
While the following review points out a number of opportunities for further development of the design and its presentation, I would like to stress I see much value in the proposed research and I believe it should be possible for the authors to address all concerns raised here. Ad: 1. The significance of the research question(s) Decision observability, the central focus of the proposed investigation, is recognized as a particularly important factor in the domain of pro-social and, more recently, pro-environmental behavior research (for specific comments on the hypotheses see the next section). A highly commendable aspect of the authors' design is the use of a psychometrically	Thank you very much for these kind words and the valuable feedback! We really appreciate this contribution to our research.	

validated and incentive-compatible task. Also, it will be useful to have evidence obtained in a pre-registered study.		
Ad: 2. The logic, rationale, and plausibility of the proposed hypotheses The main hypothesis is formulated clearly. The secondary hypotheses, while conceptually clear, could be described in more detail, with reference to variable names, the concrete operationalizations of which are to be provided elsewhere in the text (say, in the Procedures section). Testing the main hypothesis can be regarded as a conceptual replication of previous research. The novelty of the paper might thus benefit from, in addition, testing one (or more) of what is now framed as exploratory hypotheses in pre-registered a priori tests, alongside the main hypothesis. I am not saying this is necessary, but it seems like an option that should be given some thought. The scope of the proposed secondary hypotheses seems perhaps unnecessarily broad (see the next section for discussion of some other issues concerning some of these analyses). To me a paper with fewer secondary analyses and hence a more streamlined story would be just as interesting and quite possibly a nicer read.	In view of the reviews, we have decided to dispense with the secondary hypotheses and to focus entirely on the main hypothesis, for which we can guarantee to have sufficient statistical power. We agree that the test of the observability effect under our main hypothesis can be regarded as a conceptual replication of earlier work. The main contribution of our study is that this test will be conducted with more rigorous methods, involving actual observability by actual observers and a measure of actual pro-environmental behaviour. Considering the state of the literature, we reasoned that it would be most expedient to focus on this core contribution of our study. If we find an observability effect using our approach, knowledge about this effect (and data from our exploratory analyses) can be used for adequately powered, confirmatory follow-up studies on moderators and boundary conditions of the observability effect.	Deletions on pages 7 & 8

Remember that you will, for example, probably need to touch upon all the additional variables involved in the secondary tests in the exposition of the theory (especially if you actually obtain interesting results)...		
As a side note, I would recommend being somewhat more cautious in certain claims the authors make. For example, environmental behavior and people who perform it are not always perceived more positively (p. 4, lines 17-20; see e.g. Welte & Anastasio, 2010 in Environment and Behavior; Berger, 2017 in PloS ONE). Social norm interventions are not always effective or their effects may be small (p. 4, lines 26-29; see e.g. Allcott, 2011 in Journal of Public Economics). Likewise, the criticism of the subtle cues of observation approach would need to be outlined more carefully (p. 4, lines 45-47).	In response to the comment, we added some qualification to the first two statements in question, thereby also taking note of the references provided by the reviewer (many thanks for these!). With regard to the subtle cues of observation approach, we deleted the respective statement from our manuscript during the process of revising it.	Page 4: “Recent evidence suggests that individuals who comply with this norm are viewed to be more prosocial, attractive, respected, and sophisticated [17,18, but see 19]. In addition, participants who behaved more pro-environmentally have been reported to be treated more favourably in social interactions in some studies [20], but not in others [21]. Are such consequences sufficient to bring pro-environmental behaviour under the (partial) control of social norms? The success of interventions conveying normative information suggests that they are. Individuals show more pro-environmental behaviour when being informed that a majority of others approve of this behaviour or show it themselves [22–27, see also 28].” New references: 19. Welte TH, Anastasio PA. To conserve or not to conserve: is status the question?. Environment and Behavior. 2010 Nov;42(6):845-63. 20. Berger J. Signaling can increase consumers'

		willingness to pay for green products. Theoretical model and experimental evidence. Journal of Consumer Behaviour. 21. Berger J. Are luxury brand labels and “green” labels costly signals of social status? An extended replication. PloS one. 2017 Feb 7;12(2):e0170216. [...] 28. Allcott H. Social norms and energy conservation. Journal of Public Economics. 2011 Oct 1;95(9-10):1082-95. Deletions on page 5
Ad: 3. The soundness and feasibility of the methodology and analysis pipeline (including statistical power analysis where applicable) Please indicate whether the power calculations are for a one-sided or a two-sided test (p. 7, lines 30-36). Related to this, a two-sided test of the main hypothesis is proposed on p. 14 (lines 58-59), although the directional main hypothesis would seem to allow applying a one-sided test. Or are the authors unsure about the direction of the effect? In that case, this should be mentioned when presenting the hypothesis itself (p. 6, lines 31-38).	We agree that, given the state of the literature, it would be more appropriate to conduct a one-sided test of the observability effect. We now clarify this in our analysis section and also changed our power analysis accordingly.	Page 9: “This sample size allows detecting an observability effect of $d = 0.50$ with a priori power of 95% (given $\alpha = .05$, one-sided).” Page 17: “The level of significance will be set to $\alpha = .05$ (one-sided).”
In case you subsequently decide to test more than one a priori hypothesis, please indicate which, if any, corrections for testing multiple hypotheses will be employed.	As outlined above, we decided to prioritize testing one hypothesis with high statistical power and, thus, will not need to correct for multiple comparisons to control Type-I-error rate.	

If possible, I would recommend ensuring the following in the observable treatment: (a) the target knows he or she is being observed, (b) the target cannot observe the observer. Condition (b) does not seem to be satisfied in the current version of the design. To implement it, the observer can be seated 1-2 meters behind the target, for example, and observe the lights going on and off from there (the use of video could be another solution; displaying what the target sees on his or her screen on the observer's screen, while physically isolating the two individuals, could be yet another approach). Importantly, note that unless condition (b) is satisfied, it is impossible to distinguish the effect (on the target's behavior) of being observed from the effect of any number of possible subtle cues displayed by the observer, such as signs of approval or disapproval, interest or boredom, tranquility or impatience, etc. If you decide to modify the observable treatment, please make sure to modify the non-observable treatment in a corresponding manner, so that the treatments are still comparable (e.g. with respect to where participants are seated).	We thought a lot about these criteria and concluded that both of these factors (degree of knowledge/awareness/saliency of observation and observability of the observer) are likely to explain important variance in the size of observability effects. However, we are not sure that a manipulation of observability would necessarily be more valid if it satisfies these criteria; the validity depends on the operational definition of observability. Operational definitions of observability are highly variable (as evidenced by the large methodological variety in the field of prosocial behaviour), and we would find it difficult to say that one approach is universally superior. Instead, we think that observability studies can make a valuable contribution if they apply a transparent and practically or theoretically relevant operational definition of observability. We have now added a definition of observability to our introduction and we also think that it would be helpful to provide a discussion of the advantages and disadvantages of different operational definitions and manipulations. We intend to add such a discussion to the discussion section of our Stage-2-report. In this section, we will discuss, for example, the suggested possibility to arrange the positions of targets and observes in a way that allows the observer to see the target,	Page 5: “In the present study, we consider behaviour to be observable when behaving individuals as well as their behaviour can be physically observed by other individuals. Note that this definition only requires that a behaviour can be observed, not that every instance of this behaviour has to be observed. Similarly, it does not require that individuals are made explicitly aware of being observed.”
---	--	---

Concerning condition (a) above, it seems likely that in the current design some observers will not in fact engage in observation and that some targets will assume that the observers are not engaging in observation (see esp. p. 12, lines 51-59 and p. 13, lines 3-8). I realize the authors include manipulation checks that partly alleviate this concern (p. 12, lines 47-49; p. 13, lines 31-34). Still, a design where observers are explicitly tasked with observing the behavior of the targets and the targets are explicitly made aware of this would seem preferable. I am open to discussing this issue with the authors further, since I do not really understand the reasons behind the authors' design choice here.

but not vice versa. In such a situation, it would be necessary to point out to the target that there is another participant who can observe his or her behaviour. Depending on how much this information is stressed, it might put additional pressure on the target and produce experimenter demand effects. On the other hand, it might allow studying the effect of feeling observed independent of any information that is conveyed from the observer to the target. If one was interested in a very subtle type of observability, giving participants explicit information that someone else is watching them might introduce a confounder, while in other cases, it might be an essential part of the manipulation. Similarly, being exposed to the responses of potential observers could be either a confounder or a critical feature of observability.

In choosing the operational definition we chose, it was particularly important to us that the observability of behaviour varies in our study similar to how it varies in everyday situations. In some everyday situations, pro-environmental behaviour can be shown in the presence of others (who can see it and who might change their behaviour towards us as a result of what they see), but these observers are not exclusively tasked with observing us. It is not guaranteed that they will actually observe us, but if the

	situational fact that they can observe us affects our behaviour, then this could be called an effect of observability.	
An additional concern is the hypothesized treatment effect on some of the secondary outcome measures, most notably the targets' gifts to environmental organizations and the targets' gifts to their respective observers (p. 15, lines 35-40). With respect to gifts to observers, I am assuming the observer will know that the gift (which he or she will see) comes from the target sitting next to them. It seems to me that this will, critically, be the case to almost the same extent in both the observable and in the non-observable treatment (see Fig. 1 on p. 14). Thus, to postulate a treatment effect here does not seem reasonable. With respect to gifts to environmental organizations, I am assuming the donation decision will be a one-shot decision, which will be inherently very hard to observe for observers in the observable treatment (they would have to literally stare at their target's screen at the very moment the donation is made). Thus, again, to postulate a treatment effect here does not seem very reasonable.	We agree that additional explanations would be needed to justify our hypotheses about these secondary outcome measures. Perhaps even more importantly, we would require more knowledge about the observability effect in our design in general (e.g., about its existence and size) to reasonably formulate hypotheses that build on it. We thus decided not to include any hypotheses about secondary measures in our Registered Report (see above) and we also deleted the corresponding part of our analysis section. However, we still propose including these measures to generate exploratory insights into the potential boundary conditions and moderators of the observability effect.	Deletions on pages 8, 18, 19

For the same reason, one should not expect meaningful treatment differences to occur with respect to the observers' estimates of the donations made by their corresponding targets, at least assuming participants' rationality (p. 15, lines 42-45).

Perhaps some effects could occur due to biased cognition (e.g., targets might erroneously believe their donations to environmental organizations are being observed in the observable treatment, even though it seems unlikely that observing the donation would be feasible in the current design). But if the authors want to test for this type of biases, I feel that a rather specific underlying theory would need to be presented. Without having such a theory beforehand, running these additional tests will probably not benefit the paper.

Thus, the authors may want to drop these additional outcome measures and only focus on the main dependent variable. An alternative option would be to modify the procedures in such a way that treatment differences could be realistically expected to occur in case of the additional outcome measures as well (e.g., by explicitly revealing to the observer the target's

donation to the environmental organization in the observable treatment, see e.g. Vesely & Klöckner, 2018 in Journal of Environmental Psychology). I also recommend that the authors consider whether meaningful treatment differences can be expected to occur in the correlational analyses outlined on p. 15 (lines 46-59). Please keep only those tests that make sense given the issues just discussed.		
Although this is not crucial, if environmental identity is to be used as a moderator of the effect of observability (p. 6, lines 54-59), one should consider placing the environmental identity items before the PEBT task (perhaps inserting a filler task between the two), since performance of pro-environmental behavior can have short-term effects on one's environmental identity perceptions (see e.g. van der Werff et al., 2014 in Journal of Environmental Psychology).	This is a valid concern. However, such a carry-over effect might also occur when the order is reversed (indicating that one identifies as an environmentalist might lead to more pro-environmental behaviour on the PEBT). As our study focuses on the PEBT as the outcome for the confirmatory test of our core hypothesis, we decided to administer this measure first, so that it cannot be contaminated by any potential biases that might be introduced by the secondary measures. Follow-up studies based on the exploratory results of our study should, of course, take these potential order effects into account and we will also consider them in our discussion of any potential exploratory findings.	
	Reviewer 2	
Comment	Response	Change (page numbers refer to the revised version of the manuscript)
I like the idea for this study - it will be a useful	Thank you very much for these kind words and the valuable	

contribution however the results come out. Well done to the authors for creating a registered report. It is well executed. I have some design comments, and then some more minor wording ones.	feedback! We really appreciate this contribution to our research.	
It's a great design for a study, but there are one or two issues to think about. Running multiple pairs of participants at the same time in one room is elegant and efficient, but it does also mean that even in the non-observable condition, there are quite a few other people around (there is the non-observing pair partner, and also six other people in the room). So in the event of a null result, the authors may end up wondering if the T players in the non-observed condition still felt a bit observed. I would like the authors to give some rationale as to why they did not run the (to me more obvious) condition of having no-one in the room at all. At the very least I would expect to see some kind of manipulation check that participants felt less observed in the non-observed condition, and that the O players did actually see less. You could ask these questions afterwards. I was also worried that light could flare up in such a way as to still be observable in the non-observed condition (or the T players could fear that it might). In other words, some validation is needed	The question of whether the two experimental conditions differ in observability is, of course, critical to our study. The surveys to be completed by targets and observers at the end of the study include corresponding manipulation check items and in response to this comment, we added an additional question (“How well could you see what the participant next to you was doing during the experimental session?”) to the observer survey. As we have used the PEBT in our laboratory before, we are positive that the behaviour of participants in the non-observable condition is truly non-observable. The testing cubicles (which enclose the participants as well as a desk with a computer and the PEBT lights) measure 2x2x2 meters (with a 2x2 meter wall separating adjacent cubicles), and the intensity of the lights is rather moderate. We included a number of pictures below for illustration. It is true that there will be other participants in the testing room, but without drastically violating experimental instructions (e.g., standing up and wandering around) they cannot observe what other participants are doing (except for the observers that are supposed to see their	Page 14 (unchanged): “Finally, targets will be asked to rate the testing situation in our laboratory on the dimensions temperature (1 too cold – 7 too hot), lighting conditions (1 too dark – 7 too bright), and privacy/anonymity (1 very low – 7 very high). Responses to the last item will be used as a quasi-manipulation check (see below).” Page 15: “Observers will then indicate how much attention they have spent on the behaviour of the target (1 none – 7 very much), how well they could see what the target was doing during the session (1 not at all – 7 very well), and whether they know the target outside the laboratory (yes – no). These data will be used for additional robustness and manipulation checks.”

that people really do perceive themselves as less observable in the ‘non-observable’ condition. Could you provide some pilot data on this?	corresponding targets). We will monitor for these types of unlikely violations of the protocol, but have not observed them once in previous PEBT studies in the same laboratory.	
I was also not sure from the presentation how explicit it is to the participants that they will be able to finish sooner if they take the car. This is critical. For example, if Ps believed that by finishing sooner, they would just end up sitting doing nothing until the last person in the room finished the study, then they would not feel there was any real benefit to taking the car anyway, and the validity might be undermined. How will you manage people finishing at different times?	We entirely agree that this feature is crucial as well. The instructions of the PEBT version we will use for this study include the explicit sentence: “The choices you make have consequences for yourself (that is, they determine how long the experiment takes) as well as for the environment (that is, they determine how much energy is consumed during the experiment).” We now include this information in our Methods section. Data from PEBT validation studies indicate that participants take this information into account and that they are motivated to reduce the time they spend on the task. For the difficulty levels used in the proposed study, it was observed that participants are willing to take the more time-consuming bicycle option in only 30% (when the waiting-time difference between car and bicycle is 30 seconds) to 75% (when the waiting-time difference between car and bicycle is 10 seconds) of the cases (Lange, Steinke, & Dewitte, 2018). The duration of the target procedure (i.e., of the PEBT) will indeed be variable and participants can leave the laboratory as soon as they have completed all tasks (the PEBT/music rating task and	Page 11-12: “Participants directly experience two different consequences of their choice and they are explicitly informed that “the choices [they] make have consequences for [themselves] (that is, they determine how long the experiment takes) as well as for the environment (that is, they determine how much energy is consumed during the experiment).”” Page 10-11: “As soon as a participant has completed all tasks, she or he can leave the laboratory, irrespective of whether the other participant in the corresponding target-observer pair is already finished.” Page 15: “They will be presented with the first 60 seconds of nine 15 contemporary popular music pieces. [...] Completing this task is estimated to take about 25 minutes. This length is chosen to ascertain that observers will be present for the entire time targets spend on the PEBT.”

	the corresponding survey). We now state this explicitly in our Methods section. Observers are unlikely to complete all of their tasks before targets are done with the PEBT, but to be of the safe side, we prolonged the observers' music rating task to make sure that observers will be present for the whole duration of the PEBT.	
Page 3, Line 42. Sorry to be a bore, but the Homo economicus model does not require narrow self-interest. It merely requires that individuals consistently choose those options that maximise the value of their utility function. However, it is open on what the different arguments in that function are. Although many Homo economicus models assume income maximisation is what gives people utility, it is not essential to the approach to make this assumption. For example, there are 'social preference' models in which people are assumed to derive utility from the welfare of others, and these are still Homo economicus models. Best to just say 'the assumption of narrow self-interest is sufficient....'	This is a valid point indeed. We are happy to change the phrasing here.	Page 3: "Where a destination can be reached more quickly by bicycle than by car, and where switching off devices reduces carbon emissions and utility bills alike, the assumption of narrow self-interest is sufficient to account for pro-environmental behaviour."
Page 4, line 3. What is meant by 'evolved' here? Not Darwinian evolution presumably.	We did not wish to imply a specific mechanism of how group norms are established and have now replaced "evolved" by a less ambiguous alternative.	Page 4: "When a group classifies a certain behaviour as good, group members showing this behaviour are rewarded by acts of approval and affection [14, pp. 323–326]."
Page 4, line 40. What I got from the cited review on observability [25] was not	This is indeed a more accurate summary. We have adopted it	Page 5: "A recent meta-analysis revealed a small effect of

that prior findings on observability were mixed, but rather than effects were quite weak (explaining why sometimes they are statistically significant and sometimes not). Also there are various moderators. That's not the same as saying 'mixed' which implies unexplained and large heterogeneity in findings.	when discussing the results of Bradley and colleagues (2018). We also changed the abstract of our report accordingly.	observability in the broad domain of prosocial behaviour [30]. This effect was qualified by a number of moderating variables, many of which relate to the operational definition of observability. For example, observability seemed to exert a substantially stronger effect on prosocial behaviour when participants were exposed to the scrutiny of actual observers rather than to artificial cues of being watched (i.e., images of watching eyes).
Page 5, line 8. I would not say that there is a lack of conclusive evidence for observability effects; on the contrary, it's a large literature and when you meta-analyse it there is something there. Rather, I would agree that the effects people have found are typically quite weak. The limitations you mention could well explain the weakness.	At this point, we did not wish to refer to the literature on the effect of observability on prosocial behaviour (which is indeed substantial), but to the considerably smaller literature on observability and pro-environmental behaviour. What we argue is that the methods used in previous observability studies in the domain of pro-environmental behaviour did not allow for conclusive tests of the observability effect. We now clarify this in our Introduction section.	Page 6: "This lack of conclusive tests of the observability effect is a critical methodological limitation in contemporary research on pro-environmental behaviour."
Page 5, line 54. This sounds great, but can you give some more evidence for the external validity of the PEBT? i.e. does it predict some behaviour outside the lab? The sentence beginning 'Psychometric studies...' did not go on to say what these studies did or what they found.	We now elaborate on the psychometric evidence for the validity of the PEBT in our Introduction section.	Pages 6&7: "Psychometric studies revealed that the proportion of bicycle choices on the PEBT can serve as a valid, objective measure of actual pro-environmental behaviour. It is affected by variables that should theoretically affect pro-environmental behaviour (such as individual cost and environmental benefits) and related to variables that should theoretically relate

		to pro-environmental behaviour (such as environmental attitudes, concern, values, and identity). In addition, the proportion of pro-environmental PEBT choices has repeatedly been shown to correlate with self-reports of pro-environmental behaviour in everyday life [34,35].”
Page 14, Fig 1. I like the design, but I guess whether it works depends on the exact layout of the room and the furniture (T has to feel anonymous in the non-observable condition; O has to really see). Could you include some manipulation check questions about whether they felt watched, and whether O saw? Also, in the procedure, it was not clear enough to me that T knew they could get away sooner if they always took the car. If they thought that by taking the car they would just have to sit around doing nothing at the end of the session, then there would be no personal benefit to doing so. Please clarify that it is the case, and T knows it to be the case, that taking the car actually shortens the session. Also, is O’s music task a fixed duration? Or does O leave when T does? And finally, will you screen out pairs where T and O know each other?	Please see above (the first two comments of Reviewer 2) for our responses to most of the issues raised here. We included the requested manipulation check, specified that targets know about the potential to reduce the time they spend in the lab, and clarified that observers will be present for the entire PEBT procedure. With regard to the possibility that participants might know each other, we now specify in our data analysis section that these participants will not be excluded for the confirmatory analysis of the observability effect. In case of a significant finding, we will conduct robustness analyses involving this variable.	Page 17-18: “For the confirmatory test of the observability effect, we will not apply any data-related criteria for the exclusion of participants. If we find a significant effect of observability in the predicted direction, we will conduct a robustness check by repeating our analysis while excluding all participants who indicate that they know each other outside the laboratory.”
Page 15. In the secondary analyses, I was unclear whether the ANOVAs would also include	Yes, this was our idea indeed. Our apologies for not specifying this more clearly. We have now deleted these	Deletions on pages 18-19

interactions between observability and the additional variables. This would make sense to me – for example there could well be an interaction between observability and, for example, cost or benefit.	secondary analyses from our confirmatory plan. Given our sample size, tests of these interaction hypotheses will likely be underpowered. However, we can conduct exploratory analyses for potential moderators of the observability effect.	
As a final comment, if this study is illuminating I would love to see it replicated with a third condition, artificial eye cues – my prediction is that these would have some effect, but less than the real O. That would be a nice one to test.	We entirely agree that this would be an interesting comparison and we think that the opportunity to make such comparisons could be one of the important advantages of the design we proposed here. Not only could future research compare real observers versus watching eyes, but also different numbers of observers or different degrees of visibility (e.g., by making the lights more or less salient). We look forward to collecting the data on the main effect described here and then to examine potential moderators and boundary conditions in the future.	
	Reviewer 3	
Comment	Response	Change (page numbers refer to the revised version of the manuscript)
With enormous interest, I read the pre-registered report "Green when seen", submitted to RSOS. My background from Social and Environmental Psychology fits to the ideas and predictions developed in this manuscript, and I am happy to provide some feedback. Overall, I believe that this study may become an important and visible contribution to the literature	Thank you very much for these kind words and the valuable feedback! We really appreciate this contribution to our research.	

on pro-environmental behavior. I know previous work of the authors using the PEBT task, and I find it an intriguing and useful method to come as close as possible to something like "real behavior" in a lab setting. Also, the manuscript is elegantly written, considering both recent and classic relevant literature.		
1) In line 28/29 on p.3 the authors claim that "our knowledge is still limited about the factors that give rise to pro-environmental behaviour." This is somewhat trivial as this applies to virtually every scientific knowledge. I do not believe there is need for such a statement, I think stating that environmental psychology research could use more behavioral measures would be a sufficient and relevant point to make.	This was a trivial statement indeed; thank you very much for alerting us of this. We changed it accordingly.	Page 3: "Understanding the factors that give rise to pro-environmental behaviour is an important challenge for the behavioural sciences."
2) p3, l54ff: I am not yet fully convinced of the theoretical argument the authors try to make. While I applaud to link "observability" to social norms, I am not sure that the same processes act here. The authors suggest that norm-compliers being more attractive, respected, sophisticated and the such is directly linked to pro-environmental compliance under observability. Is this a logical consequence? Not for me. This needs either more lines to fully make this link clear, or some	We now elaborate on this link in our introduction section. Please note that we do not make any assumptions about the intrapsychic mechanisms underlying observability and norm effects. Instead, we argue that there is a connection on the functional level of explanation. Both the information that many people approve of pro-environmental behaviour and the observability of pro-environmental behaviour imply that individuals are more likely to receive positive social consequences when showing pro-environmental behaviour. Organisms that care	Page 4: "Are such consequences sufficient to bring pro-environmental behaviour under the (partial) control of social norms? The success of interventions conveying normative information suggests that they are. Individuals show more pro-environmental behaviour when being informed that a majority of others approve of this behaviour or show it themselves [22–27, see also 28]. In other words, pro-environmental behaviour is more likely to occur when it is made salient to

other process explanation (see 3).	about social consequences (e.g., because these are powerful secondary reinforcers) should be more likely to show a behaviour when this behaviour is more likely to result in positive social consequences.	individuals that their social environment is likely to reward such behaviour. A second prediction of a social norm account of pro-environmental relates to the role of behavioural observability. If pro-environmental behaviour is indeed driven by social consequences, it should be more prevalent in the presence vs. absence of potentially observing individuals [29]. When pro-environmental behaviour is observed, it can lead to those social consequences that reinforce its occurrence.
3) Have the authors considered "accountability" as an alternative explanation? Accountability and "being observed" are strongly intertwined and often confounded. Being held accountable for one's actions may be a particularly strong driver of pro-environmental behavior (which can be easily observed when discussing with fellow scientists about "flying").	Accountability can play an important role indeed. Rather than as alternatives, we consider these explanations as different sides of the same coin. When their behaviour is observed, people can be held accountable for it; that is, they can receive social benefits or incur social costs. We now clarify this in our introduction section.	Page 4: "When pro-environmental behaviour is observed, it can lead to those social consequences that reinforce its occurrence. When their behaviour is observed by others, individuals can be held accountable for it: they may enjoy social benefits for incurring personal cost to benefit a greater good or they may be asked to justify why they did not do so."
4) Will the pairs of observers and targets be matched, e.g. in terms of gender/age? Not because I believe these are relevant psychological constructs here, but simply because it could make a difference whether I am accountable to an in-group or an outgroup member. Also, we know from pain research that men seem to accept more/longer pain when	To address this possibility, we decided to gender-match the pairs of target and observers tested in our study. We now describe how we will do this in our Recruitment section. We do plan to match participants according to their age, because our subject pool is quite homogenous in this respect (with a large majority of participants being between 20 and 30 years of age).	Page 10: "Registration for each particular session will be open to either only male participants or only female participants, thus ensuring that all observers will be of the same gender as their corresponding targets."

females are around - similarly, they could proof to be more "environmental" when an other-sex rather than same-sex member is around.		
5) The secondary hypotheses come a bit sudden - what is the rationale for using these (cost, impact,...) and not others? I understand these are exploratory but there should be some suggestion why these should be particularly relevant.	We agree that these hypotheses and the corresponding analyses would require additional justification. In response to this comment and comments from the other reviewers, we decided to take them out and to focus on a powerful test of the main effect of observability instead.	Deletions on pages 8, 18, 19
6) Finally, for the analysis, I would suggest to think over the power analysis again. Yes, for a simple main effect the power analysis is correct, but I believe the more tests the authors make, and the more factors and covariates are involved, the less meaningful and appropriate the power analysis becomes. I do not believe that "we" need extraordinarily large samples (off the protocol: this is only one of various issues psychological research needs to address, besides better theories) but to make your research coherent, I would suggest to elaborate a little bit more here.	We entirely agree with this assessment and think that it is closely related to the issue addressed in the previous comment. We are not able to guarantee that secondary hypotheses can be tested with adequate power. For this reason, we reconsidered our approach and now only test what we can test in a robust way, that is, the main effect of observability.	Deletions of secondary/exploratory analyses from the protocol
All in all, I believe this work can make an excellent contribution, and I hope the authors perceive my suggestions as what they are: Suggestions, not criticism of their work. I wish the authors all the best with this research line.	Thank you very much again for the helpful suggestions!	

Figure of the PEBT setup (individual cubicle)

Figure of the PEBT setup (adjacent cubicles)

Figure of the PEBT setup (laboratory room)

Appendix B

Leuven, 06 February 2020

Dear Dr. Chambers,

We are pleased to submit the Stage 2 manuscript of our Registered Report titled “Green when seen? No support for an effect of observability on prosociality in an environmental conservation task” to *Royal Society Open Science*.

We confirm that we ran the experiment and analysed the data in accordance with the approved Stage 1 manuscript, uploaded as a timestamped protocol at <https://osf.io/z4jwd/>. Similarly, we confirm that we made no changes to the Introduction or Methods section of our manuscript with the exception of 1) changing future tense to past tense were required and 2) correcting typographical errors. These changes are highlighted with Track Changes in the Stage 2 manuscript. Please let us know if you prefer a different way of making these alterations transparent.

We also made a slight change to the title of our manuscript to reflect the results of our study. As indicated by the title, we did not find the predicted effect of observability on pro-environmental behaviour. Outcome-neutral tests and quality checks indicate that this null result is unlikely to be due to an ineffective manipulation of observability or arbitrary analytical choices. We conclude that the effect of observability on pro-environmental behaviour might be smaller or less general than originally expected. In light of our exploratory results, we cautiously discuss a number of factors that may account for the effect being observable in some situations, but not in others. We believe that these results and reflections can stimulate further research and contribute to an improved understanding of observability effects on prosociality in the environmental domain and beyond.

In the End section statements on page 25, we provide the link to the OSF project linked to our study. Here, we uploaded all data, study materials, and analysis scripts as well as the final version of the Stage 1 protocol. We did not collect any data prior to in-principle acceptance of this Registered Report. All data were collected between 25 October and 20 November 2019. Please let us know if you require any further information regarding our project.

Thank you very much for considering our submission.

Sincerely,

Florian Lange, Cameron Brick, and Siegfried Dewitte

Appendix C

Response to the reviewers

We would like to thank the editor and the reviewers for the careful read of our Stage 2 manuscript and their positive evaluation of our work. We also appreciate their suggestions for further improvements, which we addressed as detailed below. In general, we experienced all stages of this Registered Report to be very constructive and rewarding and we would like to thank all parties involved for their contribution.

Reviewer 1		
Comment	Response	Change (page numbers refer to the revised version of the manuscript)
When noting in the Discussion that a “validated” laboratory task was used (p. 21, line 47; authors’ pagination used throughout this report), the authors could perhaps further underscore their (exploratory) finding of choice alternatives’ costliness and environmental impact both being associated with choices, while not moderating the effect of observability on choices (p. 20, lines 26-36). This seems to support the interpretation that the absence of an effect of observability was NOT due to participants simply perceiving behavior in the PEBT task as environmentally, personally (and by extension, morally) trivial.	We agree that this is an important source of support for the validity of our approach and now stress this more explicitly.	Page 18: “These findings replicate earlier work [34] and suggest that participants take into account both these types of actual consequences when choosing between PEBT options.”
Are all unregistered exploratory statistical analyses justified, methodologically sound, and informative? Yes. However, some of these results are left without any comments from the authors (for example “Social consequences of	We added a brief comment on this section of the exploratory results. As those analyses were exploratory and did not reveal any meaningful findings, we reasoned that not much can be said with regard to, for example, the social consequences of pro-environmental behaviour. Nonetheless, we reasoned that	Page 18: “Hence, our data do not suggest that being observed showing pro-environmental behaviour relates to reputational benefits.”

pro-environmental behaviour”, see pp. 19-20).	these data will be of interest to some readers and future researchers (e.g., meta-analysts).	
Yes. The issue of relatively limited power is properly discussed (e.g. on p. 22). The authors could consider briefly discussing the possibility of conducting studies where observability is manipulated also within-subject to increase power. It could benefit other researchers to know the authors’ take on this issue.	This might indeed be a possibility to improve the statistical power of future studies. We added it to our Discussion section.	Page 20: “One possibility to mitigate this issue in future laboratory studies might be the use of within-subject manipulations of observability.”
Small issues: British vs. American English spelling is not used consistently (see e.g. p. 19, lines 56-57 vs. p. 20, lines 33-34). Not all sources are referenced consistently in the text and in the reference list (e.g., Vesely & Klöckner, 2018 appears under the number 31 on p. 22, but under the number 33 in the reference list). Some of the exploratory analyses are not at all discussed, which I personally did not mind, but it seemed a bit unorthodox.	We checked our manuscript again for occurrences of AE spelling and, in particular, changed all instances of “behavior” to “behaviour” (unless used in a proper noun). We also checked the references again and fixed the inconsistency regarding the paper by Vesely & Klöckner. Finally, we added brief discussions of our exploratory findings where those had been missing.	Page 18: “exerted marked effects on PEBT choice behaviour” Page 17: “Taken together, these results do not provide support for an effect of our observability manipulation on any of the donation-related variables listed above.” Page 18: “Hence, our data do not suggest that being observed showing pro-environmental behaviour relates to reputational benefits.”
Reviewer 2		
Comment	Response	Change (page numbers refer to the revised version of the manuscript)
- Will the paper show somewhere that this was a registered report (e.g. in the title or heading or intro?) I think this is a real strength and readers should know.	We agree and added this information to both the title and the abstract of our manuscript.	Page 1: “Green when seen? No support for an effect of observability on prosociality in an environmental conservation task: a Registered Report” Page 2:

		“In the present Registered Report, we used a recently validated laboratory procedure of repeated dilemmas to test whether the presence of actual observers affects costly prosocial behaviour in the domain of environmental conservation.”
- Page 20, line 35. I take it the F-ratios referred to here are for the interaction. I would prefer to see them all given in full with their degrees of freedom.	This is correct. We added the full results for both interaction effects to clarify this point.	Page 18: “Neither cost, $F(2.18, 379.49) = 1.01, \eta p^2 = .01, 95\% \text{ CI } [.00, .03]$, nor impact, $F(1, 174) = 0.94, \eta p^2 = .01, 95\% \text{ CI } [.00, .05]$, moderated the effect of observability.”
- Page 20, line 53. I was a bit surprised by this mediation analysis, since in a sense there is nothing to mediate. But it is fine to report it. However for what it is worth there is a significant indirect effect, in the absence of any direct one. What does this mean? Is this worth discussing more in the discussion, and perhaps explaining a bit more here in the results.	In response to this comment, we decided to delete this exploratory analysis because 1) we did not want to let further elaborations on its results dominate our discussion and 2) we thought that it did not inform the interpretation of our results over and above the correlational findings we report.